# The actin binding sites of talin have both distinct and complementary roles in cell-ECM adhesion

**Darius Camp**[1], **Bhavya Venkatesh**[1], **Veronika Solianova**[1], **Lorena Varela**[2], **Benjamin T. Goult**[2,3], **Guy Tanentzapf**[1] *

**1** Department of Cellular and Physiological Sciences, University of British Columbia, Vancouver, British Columbia, Canada, **2** School of Biosciences, University of Kent, Canterbury, Kent, United Kingdom, **3** Department of Biochemistry, Cell & Systems Biology, Institute of Systems, Molecular & Integrative Biology, University of Liverpool, Crown Street, Liverpool L69 7ZB, United Kingdom.

* tanentz@mail.ubc.ca

**Data Availability Statement:** All relevant data are within the manuscript and its Supporting Information files.

## Abstract

Cell adhesion requires linkage of transmembrane receptors to the cytoskeleton through intermediary linker proteins. Integrin-based adhesion to the extracellular matrix (ECM) involves large adhesion complexes that contain multiple cytoskeletal adapters that connect to the actin cytoskeleton. Many of these adapters, including the essential cytoskeletal linker Talin, have been shown to contain multiple actin-binding sites (ABSs) within a single protein. To investigate the possible role of having such a variety of ways of linking integrins to the cytoskeleton, we generated mutations in multiple actin binding sites in *Drosophila* talin. Using this approach, we have been able to show that different actin-binding sites in talin have both unique and complementary roles in integrin-mediated adhesion. Specifically, mutations in either the C-terminal ABS3 or the centrally located ABS2 result in lethality showing that they have unique and non-redundant function in some contexts. On the other hand, flies simultaneously expressing both the ABS2 and ABS3 mutants exhibit a milder phenotype than either mutant by itself, suggesting overlap in function in other contexts. Detailed phenotypic analysis of ABS mutants elucidated the unique roles of the talin ABSs during embryonic development as well as provided support for the hypothesis that talin acts as a dimer in *in vivo* contexts. Overall, our work highlights how the ability of adhesion complexes to link to the cytoskeleton in multiple ways provides redundancy, and consequently robustness, but also allows a capacity for functional specialization.

## Author summary

The construction of tissues in animals requires cells to form attachments (adhesions), to other cells or the surrounding mixture of proteins and molecules known as the Extracellular matrix (ECM). Cell adhesion to the ECM has an important role in providing the mechanical stability that allows cells to arrange into different shapes during animal development. Key to this stability is their ability to connect to and organise the intracellular

**Funding:** Funding for this study was provided by a grant to GT from the Natural Sciences and Engineering Research Council of Canada (NSERC, RGPIN-2018-04648; https://www.nserc-crsng.gc.ca/index_eng.asp), the Canadian Institute for Health Research (CIHR, AWD-026167 CIHR 2023; https://cihr-irsc.gc.ca/e/193.html) and to BTG from the Biotechnology and Biological Sciences Research Council (BB/S007245/1). The funders had no role in study design, data collection and analysis, decision to publish, or preparation of the manuscript.

**Competing interests:** The authors have declared that no competing interests exist.

cytoskeleton such that it forms a scaffold that is both strong and flexible. To study the strategies used by cells to link the ECM to the cytoskeleton we introduced mutations into a protein called talin that is a key intermediary in adhesions between the outside environment and the cytoskeleton. Talin has different ways of linking to the actin cytoskeleton as it contains multiple domains that bind actin. Using a genetic approach, we disrupted these domains separately or together to block actin binding. We found these domains have both unique and complementary roles in cell-ECM adhesion indicating that having multiple actin-binding sites provides functional redundancy and specialization. Overall, our findings provide mechanistic insight into how cells can regulate cell adhesions by changing how they link to the cytoskeleton.

## Introduction

A central feature of cell adhesion complexes is their ability to couple the inside and outside environments of the cell. Transmembrane adhesion receptors bridge the inside and outside environments, and recruit and assemble multi protein complexes inside the cell to control the intracellular environment. A key function of the cytoplasmic adhesion complex is to link the transmembrane adhesion receptor to the cytoskeleton. For this reason, multiple proteins with actin binding activity are found within adhesion complexes. For example, cadherin-based adherens junctions may contain the actin binding proteins α-catenin, vinculin, filamin, and afadin [1–5]. Particularly notable in this regard are integrin-based cell-ECM junctions which may contain dozens of proteins with actin binding activity such as talin, vinculin, filamin, α-actinin, arp2/3, Zyxin, Vasp, and tensin [6–14]. Moreover, some of the actin linkers that associate with the integrin adhesion complex contain several actin binding domains, for example both tensin and talin contain at least 3 known actin-binding sites [6, 14]. This raises the question of why cell adhesion complexes and in particular integrin-based adhesions require so many different ways of connecting to actin.

Talin has been studied extensively for its role as an essential linker between integrins and the actin cytoskeleton [15,16]. Talin is a large (~270 kDa) cytoplasmic protein that, in addition to having three potential actin-binding sites, also has binding sites for many other partners, including some, most notably vinculin, that can also bind actin directly [17–19]. The ability of talin to support direct linkage to actin has been the subject of extensive study in *in vitro* and cell culture systems [6,20]. The three potential actin-binding sites (ABSs) of talin are located at its N-terminal region (ABS1) [21], at the center of its long rod domain (ABS2) [22,23], and near the end of the C-terminal region (ABS3) [24,25]. The best characterized Talin ABS is the highly conserved C-terminal ABS3 also known as the talin-HIP1/R/Sla2p actin-tethering C-terminal homology or THATCH domain [24–26]. Both ABS2 and ABS3 are located within the Talin rod, a region which is made up of 13 helical bundles (labeled R1-R13), that are followed by a dimerization domain (Talin-DD, Fig 1A). The repeating helical bundles of the rod give Talin one of its most prominent features, its ability to undergo rapid conformational changes in response to mechanical force [27,28]. According to current models, the helical bundles unfold when they are exposed to mechanical stretching forces and refold when these forces are reduced [27–29]. Unfolding of the helical bundles exposes cryptic protein binding domains thereby changing the function of talin [27–29]. The unfolding of talin occurs in a stereotypic fashion meaning that as force increases specific helical bundles are impacted. Talin can therefore act as a mechanosensor, where specific levels of mechanical force drive certain outcomes [30].

A number of studies have provided insight into the possible unique function of each of the talin ABS [23,31]. ABS3 is hypothesized to be the initial site to engage actin and allow the

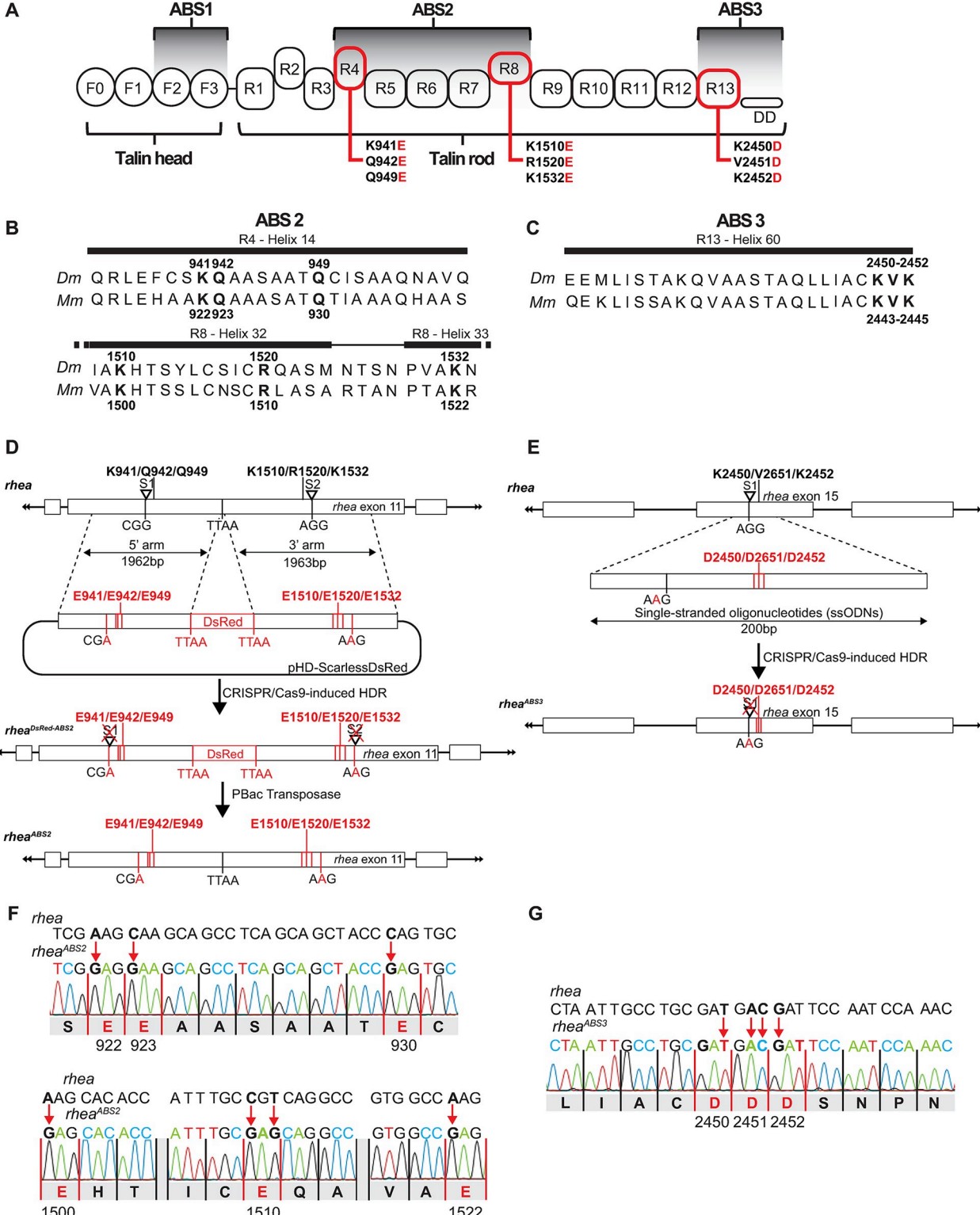

**Fig 1. Generating talin ABS2 & ABS3 mutants.** (A) Schematic representation of the head/FERM (F0-F3) and rod domains (R1-R13) of Talin along with the actin binding regions 1, 2 and 3 (ABS1, ABS2 & ABS3) and the Dimerization Domain (DD). ABS2 and 3 were respectively disrupted by 6 and 3 amino acid substitution in R4 & R8 and R13 domains (shown in red). (B, C) Sequence alignment of the ABS2 (B) and ABS3 (C) regions from fly (Dm Talin) and mouse (Mm talin 1). The mutated amino acids (in bold) are conserved in both species. (D, E) CRISPR/Cas9 genome editing strategy to generate ABS2 and ABS3 mutations by homology-dependent repair (HDR). ABS2 mutations (amino acids in red) were introduced using

2 guide RNAs and a dsDNA plasmid donor containing a Scarless-DsRed cassette to facilitate genetic screening. The DsRed cassette was inserted at a native TTAA site to allow for scarless removal of the DsRed cassette by Pbac transposase (D). A single stranded oligonucleotides (ssODNs) donor vector and 1 guide RNA were used to obtain the ABS3 mutation (amino acids in red) (E). Cut sites targeted by Cas9 (labeled S1 & 2 in D and E) were made inactive by silently mutating the associated PAM sequence required for Cas nuclease activity (3 nucleotides shown on wildtype *rhea* locus, mutated nucleotide shown in red mutated locus) in both donors to block CRISPR from cutting the newly inserted target sequences. (F, G) Representative electropherogram of ABS2 (F) and ABS3 (G) mutant flies confirming the presence of the mutations.

application of mechanical force across the talin molecule causing it to stretch and reveal cryptic protein binding sites. One of these cryptic binding sites is ABS2, which has low affinity for actin in the absence of applied mechanical force [23]. In turn, ABS2 is thought to facilitate the formation of a strong high-mechanical force bearing actin linkage [23,31]. The role of ABS1 is less clear as it overlaps with many other domains that are essential for talin function, most notably a key integrin-binding site [21,32]. It is unclear whether the ABS1 domain can bind actin if talin is bound to integrin but biochemical studies have provided evidence suggesting that it can [33]. Importantly, data shows that unlike ABS2 and ABS3, the ABS1 has an inhibitory effect on actin assembly by suppressing barbed-end elongation of actin filaments [33]. Presently, the relative importance of each talin ABS is not known, nor do we know how the function of the different ABS domains is coordinated.

*Drosophila* has proven to be a powerful model to elucidate the mechanisms driving assembly and regulation of integrin-based cell-ECM adhesions in a genetically tractable *in vivo* system. Loss of integrin-mediated adhesion impacts many aspects of fly development and tissue homeostasis which means adhesion phenotypes can be studied in diverse contexts. These contexts include those that involve dynamic short-term adhesions such as those that occur during embryonic dorsal closure in the fly embryo [34–36], or stable long-lasting adhesion such as those that occur during muscle-tendon attachment in larva [37,38]. The well conserved, yet simpler, integrin adhesion complex of flies facilitates these studies as it contains less redundancy, for example flies have only one talin (encoded by *rhea* in flies) and two β-integrin subunits, compared to two talin genes and 8 β-integrin subunits in vertebrates [39]. *Drosophila* has proven to be an especially good system in which to carry out structure-function analysis of components of the integrin adhesion complex [17,40–46]. The relative ease and speed of generating and characterizing point mutations in components of the adhesion complex, the wide availability of various markers for integrins and their associated proteins, the sophisticated methodology to carry out quantitative phenotypic analysis, all facilitate structure-function studies.

Here we describe the results of a systematic genetic analysis of the function of ABS2 and ABS3 *in vivo*. Our results show that both ABS2 and ABS3 are required for proper Cell-ECM adhesion during development, suggesting that their functions are unique and non-redundant in many contexts. In particular we find a requirement for ABS2 in the initial localization of talin to sites of adhesion and the assembly of Cell-ECM junctions. In comparison, ABS3 mutants are recruited to sites of adhesion where they assemble adhesions, but exhibit a variety of adhesion defects. Nonetheless, we also provide evidence that, at least in some contexts, ABS2 and ABS3 act redundantly and overlap in function. Together, these findings provide a mechanistic framework for understanding the role of direct binding of actin by talin and suggest a possible explanation for why cell adhesion complexes use multiple strategies to link to actin.

## Results

### Selection of mutations to disrupt the talin ABS domains

Several biochemical studies have defined and characterized the ABS2 and ABS3 domains of Talin [17,22–24,31]. The ABS2 domain extends over a sizable region of the Talin Rod, over

600 amino acids long, as it involves two different helical bundles, R4 and R8, that are separated by 3 domains and ~650 amino acids (Fig 1A) [6,31]. The R4 and R8 region share some similarity with R13 which encompasses ABS3 and contain a series of highly conserved surface residues that were identified as important for binding to actin. Mutating three of these conserved surface residues in R4 (K922, Q923, Q930 in vertebrate talin), and three additional residues in R8 (K1500, R1510, and K1522 in vertebrate talin) to glutamic acid residues reduced the actin binding ability of talin, as judged by *in vitro* actin sedimentation assays, by over 60% [31]. These sites are conserved in *Drosophila* and correspond to K941, Q942, Q949 in R4 and K1510, R1520, K1532 in R8 of *Drosophila* talin, respectively (Fig 1B and 1C). To disrupt ABS2 we mutated these 6 residues to glutamic acid in *Drosophila* talin and confirmed using a series of biochemical experiments that introducing these mutations did not impact protein folding or stability. To disrupt ABS3 the conserved residues K2450, V2451, K2452 were mutated to aspartic acid, a change that has been shown by us and others to greatly reduce, though probably not eliminate, actin binding through ABS3 but not impact protein folding or stability [17,23,24,31]. Previous analysis of the KVK/DDD ABS3 mutants [17,24] found anywhere between 40%-73% decrease in actin binding in an *in vitro* actin co-sedimentation assay. These *in vitro* experiments likely represent an overestimate of the actin binding activity of these mutations as they are designed to optimize actin binding. To test the structural integrity of the mutated domains using Nuclear Magnetic Resonance (NMR) and Circular Dichroism (CD), the wildtype R4, R8 and R13 domains, and the corresponding mutated R4 domain (R4_3E; K941, Q942, Q949), R7/R8 domains (R7R8_3E; K1510, R1520, K1532) and R13 domain (R13_3E; K2650, V2651, K2652) were each recombinantly expressed and purified (see materials and methods). The 1H,15N HSQC spectra of all three wildtype domains show good dispersion and uniform, peak distribution confirming that the fly rod domains are correctly folded (S1A Fig). Similarly, the CD spectra confirmed that all three rod domains were helical in nature (S1D–S1F Fig). The mutant proteins all gave HSQC and CD spectra that were similar to the corresponding wildtype versions confirming that the mutations to the exposed surface residues did not perturb domain folding or stability, similar to what was shown previously for equivalent mutations in mouse talin [22, 31]. Having validated the mutants did not affect the structural integrity of the fly Talin domains, we next generated fly mutant lines containing the 6 point mutations in ABS2 (K941E, Q942E, Q949E, K1510E, R1520E, K1532E), referred to henceforth as *rhea*^ABS2^ allele or ABS2 mutant and the 3 point mutations in ABS3 (K2450D, V2451D, K2452D), referred to henceforth as *rhea*^ABS3^ allele or ABS3 mutant (see materials and methods, Fig 1A–1C).

## Generation of ABS mutations using CRISPR

To introduce the ABS2 and ABS3 mutations into *rhea*, the gene that encodes talin in flies, we used a CRISPR/Cas9-based approach. Two different CRISPR/Cas9-mediated methodologies were used to genetically engineer the endogenous *rhea* locus, a single-stranded oligodeoxynucleotides (ssODNs) approach was used to introduce mutations into the ABS3 domain while a strategy of "scarless" homology-directed repair (HDR) with a double-stranded DNA (dsDNA) donor template (see materials and methods, [47]) was used to alter the ABS2 domain (Fig 1D and 1E). Briefly, to modify the ABS2 sequence two guide RNAs (gRNAs) that targeted sites 5' and 3' of the sequence corresponding to the ABS2 domain were used as well as a dsDNA donor construct. The donor construct featured homology arms containing approximately 2kb of sequence flanking a visible selection marker, DsRed, that contained PBac transposase target sites allowing excision (see materials and methods, Fig 1D). Each of the two homology arms contained a cluster of mutations in the ABS2 site, namely K940D/Q941D/Q948D in R4 and

K1510E/R1520E/K1532E in R8. In the first step of mutagenesis the DsRed marker was used to identify successful editing events while in the second step this DsRed marker was removed by crossing to a PBac transposase source. To modify the ABS3, a ssODN donor of 200 base pairs long was synthesized that contained the K2670D/V2671D/K2672D mutations (Fig 1E). This was followed by a genetic and PCR based screening approach to identify candidate mutant lines (see materials and methods). Following the identification of multiple candidates, ABS2 and ABS3 mutant lines were sequenced extensively both upstream and downstream of the mutation sites to confirm the presence of the mutation and to ensure no other deleterious events took place at the *rhea* locus (Fig 1F and 1G). Using the approach detailed here we successfully generated multiple independent mutated lines of ABS2 and ABS3. As the different ABS2 and ABS3 mutant lines displayed identical sequencing results, phenotypic analysis was then performed on only 1 line per ABS mutation to investigate their role.

## Actin binding through the talin ABSs is required for viability

We noted that the ABS2 ($rhea^{ABS2/ABS2}$) or ABS3 ($rhea^{ABS3/ABS3}$) mutant flies were non-viable and failed to complement null alleles of *rhea*. Moreover, introducing a ubiquitously expressing talin rescue construct (ubi::TalinGFP, [48]) into the background of null alleles of *rhea*, or homozygous mutants of the ABS2 or ABS3 mutant alleles rescued the embryonic lethality associated with loss of talin to the same extent, specifically, until pupation (Fig 2A). These data are consistent with ABS2 or ABS3 mutant alleles acting like genetic loss-of-function alleles of *rhea*. Further analysis of the stage at which lethality took place in the different mutants showed that while homozygous null mutants of *rhea* or homozygous ABS2 mutant flies died during the first instar larval stage, homozygous ABS3 mutants died in the second instar larval stage and displayed significantly higher L1 survival rate than ABS2 mutants (Fig 2A and 2B). This suggested that the phenotype of the ABS2 mutant allele was similar to that of a null allele of *rhea* while the phenotype of the ABS3 mutant allele was not as strong. Importantly, when we generated transheterozygotes, flies that contain both the ABS2 and ABS3 mutant alleles but do not produce wildtype talin ($rhea^{ABS2/ABS3}$) (see materials and methods), the flies exhibited a much weaker phenotype than either ABS mutants by themselves (Fig 2A and 2B). Specifically, ABS2/ABS3 transheterozygote mutant flies survived until pupal stages similarly to talin mutants expressing the talin rescue construct. To learn more, we employed a more sensitive method to compare the strength of the phenotypes of talin mutants. In particular, we measured "motility" by quantifying the percentage of fly larvae that were able to move more than 5 mm over the course of at least 1 hour, (Fig 2C–2H see materials and methods). This analysis showed that null mutant flies which lack all talin function exhibit almost no movement, with only about 4% of larva moving at least 5 mm during the given time period compared to 100% of wildtype flies. Flies homozygous for the ABS2 mutant allele exhibit a somewhat weaker phenotype as 25% moved at least 5 mm, while for the ABS3 mutant that proportion increased to 72%. In comparison nearly 88% of ABS2/ABS3 transheterozygotes flies exhibited such motility. To determine whether these mobility phenotypes corresponded to visible defects in muscles we introduced actinin::GFP, a marker for sarcomeric architecture and muscle attachment, into the background of our mutants (Fig 2I–2L). Mutants were analyzed in the latest larval stage that they were able to reach, and although null alleles of talin and ABS2 mutants proved too fragile for the staining protocol, we were able to image second instar larva from ABS3 mutants and third instar larva from ABS2/ABS3 transheterozygotes mutants. Surprisingly, we observed no noticeable muscle defects in ABS3 mutants or ABS2/ABS3 transheterozygotes suggesting alternative causes for their motility phenotype, such as neuronal dysfunction. These data are consistent with the interpretation that the ABS2 mutant allele is a strong loss of function allele

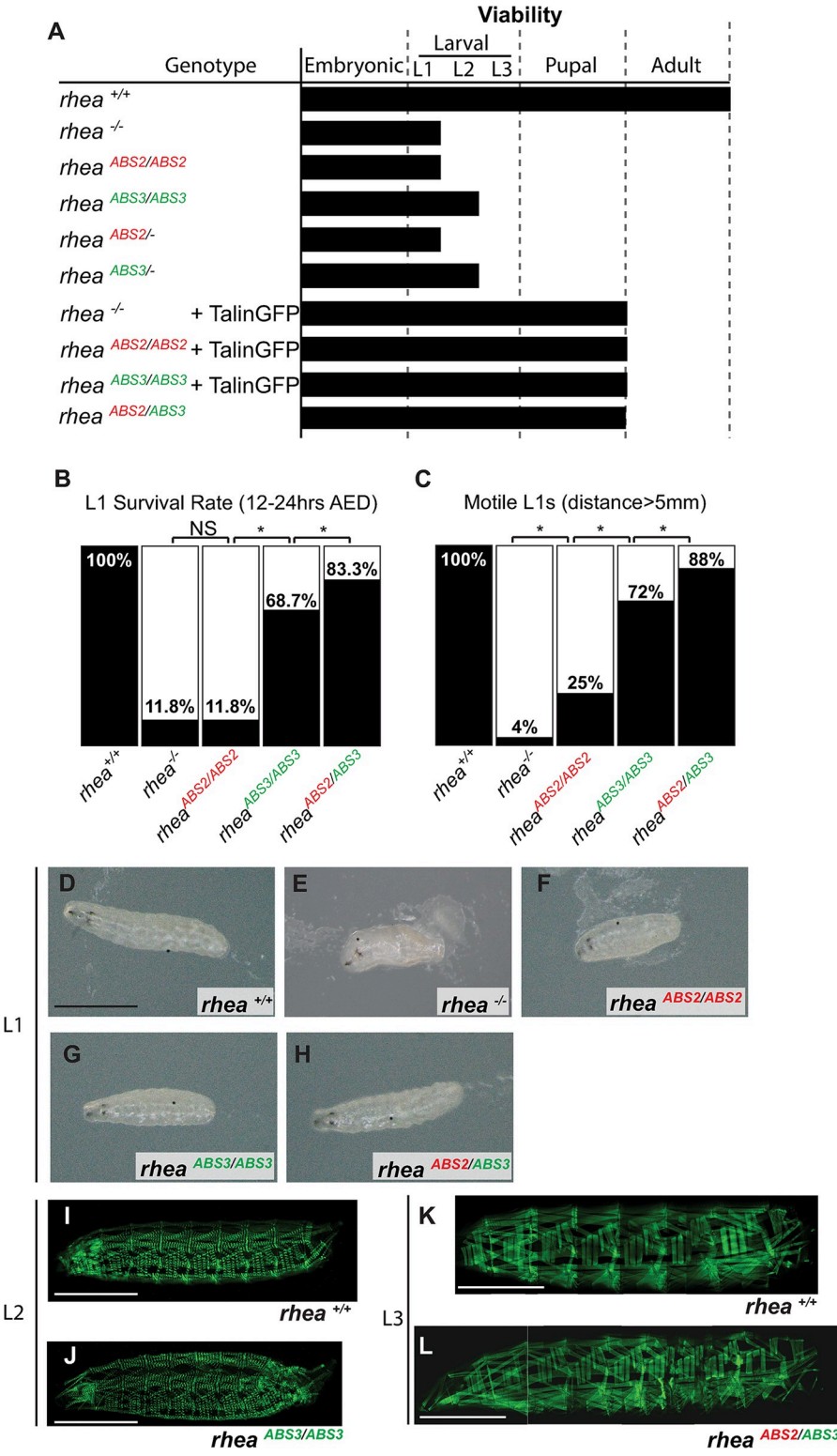

**Fig 2. Viability and motility of talin ABS2 and ABS3 mutants.** (A) Viability of ABS2 and ABS3 homozygote mutants (*rhea*$^{ABS2/ABS2}$ and *rhea*$^{ABS3/ABS3}$ respectively) compared to wildtype (Wt) (*rhea*$^{+/+}$) and null (*rhea*$^{-/-}$) flies. Talin null allele does not complement either ABS mutants but transheterozygote (*rhea*$^{ABS2/ABS3}$) flies display longer viability similar to mutant flies rescued by the expression of a Wt talin (+TalinGFP). (B) L1 Survival rate. Percentage of L1 larvae successfully hatching 12–24 hours after egg deposition (AED). (C) L1 motility. Percentage of L1 larvae capable

of moving at least 5 mm away from hatching site. (D-H) Representative picture of L1 larvae from each genotype. (I-L) Confocal images of whole mount L2 (I, J) and L3 (K, L) larvae of the indicated genotype expressing α-actinin-GFP to visualize muscle structure. Survival and motility rates compared using chi-square test: * $P<0.05$; NS, no significance; all genotypes n>100. Scale bars: 1 mm.

that is only slightly less strong than a complete null, the phenotype of the ABS3 mutant allele is intermediate, while ABS2/ABS3 transheterozygotes exhibit only a mild phenotype.

## The talin ABSs are required for multiple aspects of embryonic development

Complete loss of talin in embryos, achieved by using genetic tools to remove both the maternal as well as zygotic expression of talin (see materials and methods), results in strong embryonic defects that impact multiple tissues and processes [15]. Notable among these are disruption of long-lasting stable attachments of muscle cells to tendon cells, and the failure of two dynamic morphogenetic processes, germband retraction (GBR) and dorsal closure (DC). The attachment of muscles to tendons occurs via integrin-based adhesions at myotendinous junctions (MTJs) that form in late fly embryogenesis [49]. Germband retraction involves the pulling back of the caudal end of the embryo, which at stage 12 is looped back so that it is near the head, such that it reaches its final position at the posterior of the embryo [50]. Dorsal Closure is a morphogenetic process in which a large hole on the dorsal side of the embryo is enclosed by the movement of two lateral sheets of epithelia [36,50]. Fly embryos that completely lack talin ($rhea^{-/-}$) are characterized by a high proportion of embryos that exhibit an abnormal folded appearance due to failed GBR, a high proportion of embryos that exhibit a large dorsal hole due to failed DC, and a large proportion of embryonic muscles that are detached and rounded up due to failed attachment of muscles to MTJs (Fig 3A and 3B; quantified in I-K). Flies that completely lack wildtype talin but contain the talin ABS2 mutant allele exhibited a high penetrance of DC and GBR defects, similar to the talin null allele, but showed a lower proportion of detached muscles compared to the null allele (Fig 3C; quantified in 3I–3K). Flies that completely lack wildtype talin but contain the ABS3 mutant allele also exhibited a high penetrance of DC and GBR defects, similar to the null allele, but a much lower proportion of detached muscles compared to the null allele (Fig 3D; quantified in 3I–3K). In contrast flies that completely lack wildtype talin but contain both the ABS2 and ABS3 mutant alleles exhibited a much lower penetrance of DC and GBR defects relative to the null allele, as well as a much lower proportion of detached muscles (Fig 3E; quantified in 3I–3K). Assaying a non-embryonic integrin-associated phenotype, the formation of blisters in the fly wings due to loss of apposition of the two layers of the fly wing, revealed similarly strong phenotype in $rhea$ null, ABS2, and ABS3 mosaic mutants (Fig 3H, see materials and methods). Taken together these data show that the talin ABS sites play diverse, essential roles in fly development and tissue maintenance, and, as revealed by the transheterozygotes, they share a great deal of functional redundancy. Nevertheless, in some processes, such as in muscle attachment, disruption of ABS2 causes stronger phenotype than disruption of ABS3, suggesting it has a broader role.

## ABS2 but not ABS3 is required for talin recruitment to sites of adhesion

The null-like embryonic phenotypes observed in flies expressing only the ABS2 mutant allele were consistent with a strong disruption to integrin-mediated cell-ECM adhesion. Such a defect could be caused by failure to express talin. To test if there were differences in the expression of the ABS mutations we carried out qPCR experiments which showed that both the ABS2 and ABS3 mutant alleles were expressed at similar levels to that of wildtype talin (Fig 4L). Since qPCR measures transcription but not translation or protein stability we used

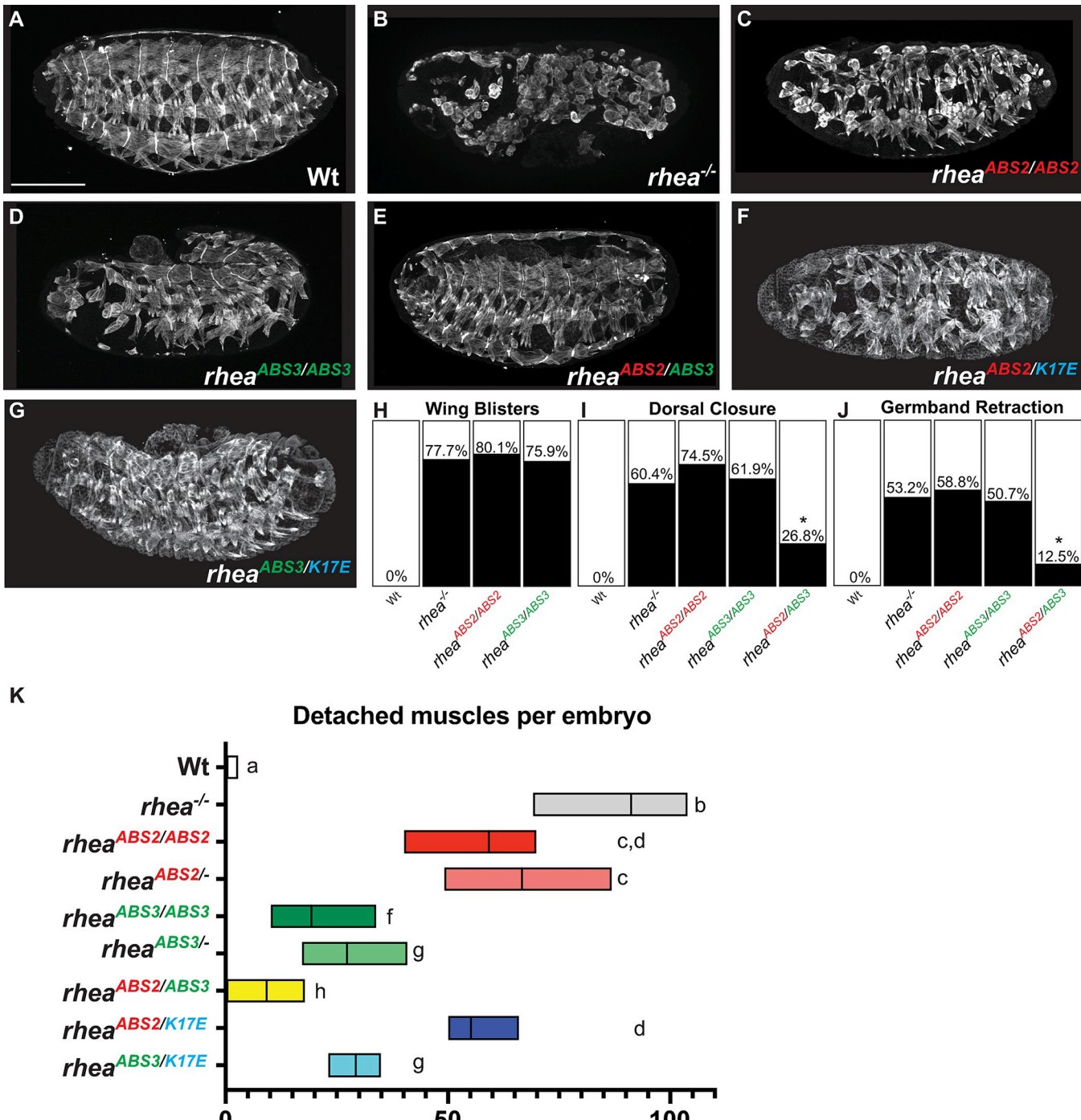

**Fig 3. Talin ABS requirement for muscle and embryonic development.** (A-G) Confocal images of whole-mount embryos stained with muscle cytoskeleton marker Myosin Heavy Chain (MHC) at stage 17 (scale bar: 50 μm). (H-J) Penetrance of wing blisters (H), dorsal closure (I) and germband retraction (J) defects in embryos of the indicated genotypes. Dorsal closure and germband retraction defects of $rhea^{ABS2/ABS3}$ compared to $rhea^{ABS3/ABS3}$ using chi-square test: *$P<0.05$; all genotypes n>30. (K) Number of detached muscles per embryo. Floating bar chart representing minimum and maximum value per genotype (n>10). Mean value represented by a solid line. Each lower-case letter (a-g) indicates a different statistical group where two different letters represent statistically different results ($P<0.05$) using one-way ANOVA, Dunnett's post-hoc, $P<0.05$.

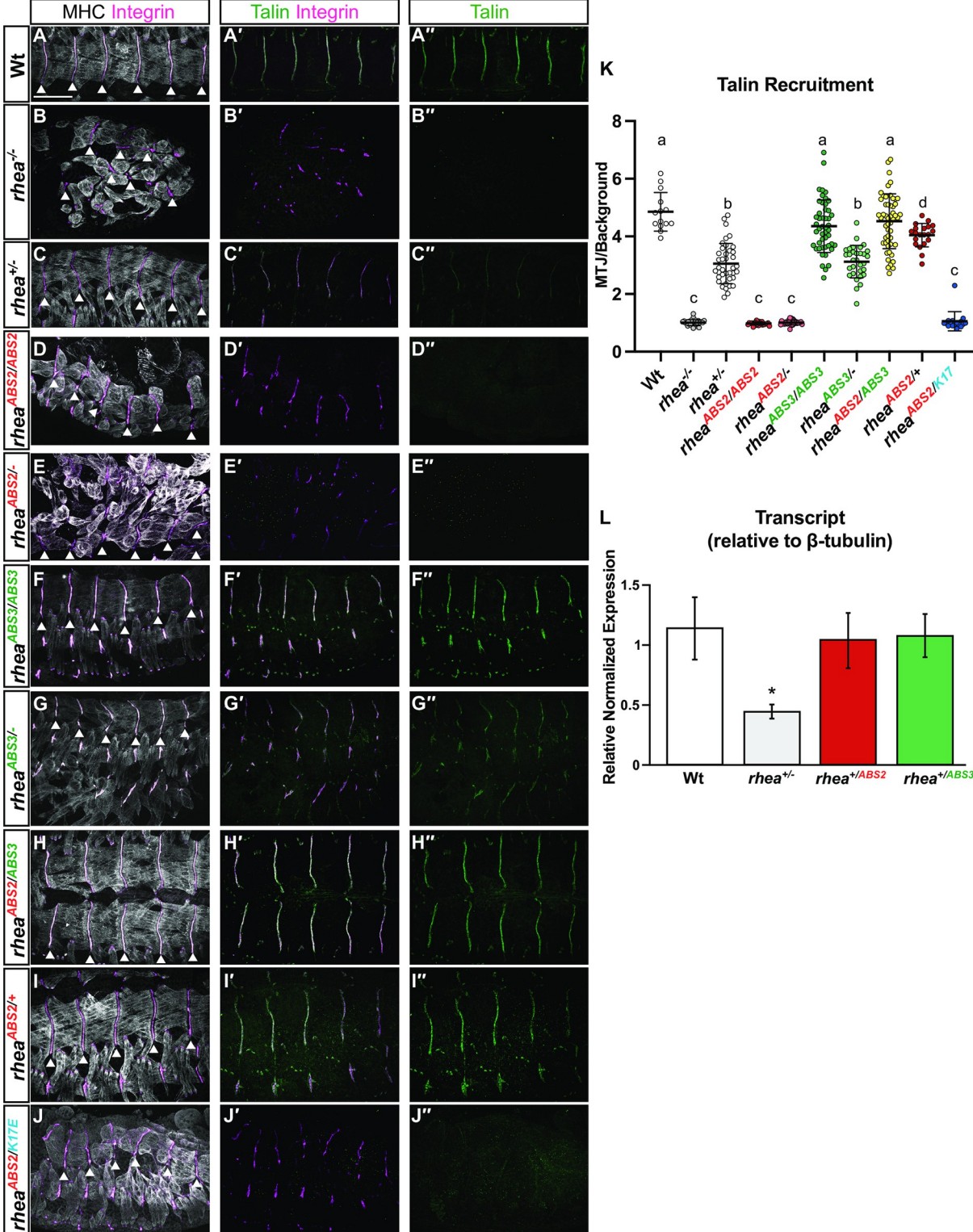

**Fig 4. ABS2, but not ABS3 is required for talin recruitment to sites of adhesion.** (A-J′′) Talin recruitment at MTJs. Confocal z-stacks of MTJs and surrounding muscles in stage 17 embryos stained for MHC (white), the integrin subunit αPS2 (magenta) and Talin (green). Scale bar: 25 μm. MTJs location was determined by αPS2 staining in between muscle cells (arrowheads in A-I). (K) Relative localization of Talin to MTJs. Mean value represented by solid line, error bar show s.e.m. and dashed line indicates the absence of recruitment (n>15 per genotype). Each lower-case letter (a-c) indicates a different statistical group (*P*<0.05) using one-way ANOVA, Dunnett's post-hoc. (L) qPCR analysis of

*rhea*^+/−^, *rhea*^+/ABS2^ and *rhea*^+/ABS3^ heterozygote flies compared to wildtype (Wt) flies (two-tailed Student's t-test, *$P<0.05$). Error bars represent s.e.m.

immunohistochemistry to analyze Talin protein levels at sites of adhesion in MTJs (quantified in Fig 4K). To assess the ability of the ABS2 mutant alleles to support recruitment and/or maintenance of talin to MTJs compared to the null and ABS3 mutant alleles of talin we took advantage of a previously described methodology (49). In this approach integrin, which localises to muscle ends independently of talin, is used to label MTJs allowing the determination of the ratio of talin found at MTJ compared to talin found in the cytoplasm (Fig 4A) (49). In a wildtype embryonic muscle, talin is enriched at MTJs by approximately five-fold compared to the cytoplasm (Fig 4K). In embryos lacking any talin expression (*rhea*^-/-^), muscles detached from the ECM but MTJs were still identifiable by integrin labeling allowing us to observe that talin was not enriched at MTJs (Fig 4B, the 1:1 ratio of intensity seen in Fig 4K is consistent with a lack of talin enrichment). In heterozygous embryos that contained only one copy of the wildtype talin allele (*rhea*^-/+^) levels of talin were reduced by about half in MTJs compared to fully wildtype controls (*rhea*^+/+^) (Fig 4C and 4K). In embryos that expressed only the ABS2 mutant allele (*rhea*^ABS2/ABS2^) muscle detachment was observed and, as in the null allele, integrin but not talin was recruited to MTJs (Fig 4D, and 4K). Finally, in embryos that expressed only the ABS3 mutant allele (*rhea*^ABS3/ABS3^), talin was recruited to MTJs to a similar degree as in wildtype controls (Fig 4F and 4K). Overall, these results suggest that the ABS3 domain of talin is dispensable, while the ABS2 domain of talin is at least partially required, for talin recruitment to, or maintenance at, sites of adhesion. These results however left open the possibility that the ABS2 allele interfered with talin expression by impinging on the translation or stability of the talin protein.

Further insight into the question of whether the ABS2 allele interfered with protein expression emerged in experiments that measured talin levels at MTJ when the mutant ABS3 allele was combined either with a null or an ABS2 allele. In a mutant talin ABS3 heterozygotes, meaning a background that contains a talin null allele and an ABS3 mutant allele (*rhea*^ABS3/-^) the recruitment of the ABS3 mutant talin protein to sites of adhesion at MTJs was reduced by half compared to a wildtype (*rhea*^+/+^) (Fig 4G and 4K). This is similar to what is observed in heterozygous null embryos that have one copy of wildtype talin and one copy of a null allele of talin (*rhea*^+/-^; Fig 4C and 4K). Such results were consistent with the idea that the mutant talin ABS3 protein was recruited with similar efficiency as the wildtype talin protein to MTJs. Importantly, in a background that contained one copy of the ABS3 and one copy of the ABS2 mutant alleles (*rhea*^ABS2/ABS3^) the levels of talin recruited to MTJs were similar to those seen in wildtype controls (Fig 4H and 4K). These results are consistent with the interpretation that the talin protein with the ABS2 mutation is abundantly expressed and can be visualized at the MTJs if it receives assistance in reaching the membrane, as it does in the presence of ABS3.

We sought to test the idea that a main causative defect in ABS2 mutants was their failure to localize to, or be retained at, the membrane, and that this defect could be overcome by synergistic interactions with ABS3 mutants [51]. To this end we used an allele of talin, *rhea*^K17E^, that contains a point mutation that interferes with the direct binding of talin to Rap1, an interaction that we showed previously does not impact the expression of talin but disrupts the proper recruitment and/or retention of talin to the membrane [43]. Consequently, homozygous *rhea*^K17E^ mutants behave like talin null mutants, even though the talin protein is made in wildtype amounts. Our hypothesis predicted that, unlike ABS3 mutants, *rhea*^K17E^ and *rhea*^ABS2^ mutants share a molecular defect in terms of recruitment. For example, as we previously showed for *rhea*^K17E^ heterozygotes (*rhea*^K17E/+^) [43], in an embryo containing one copy of

wildtype talin and one copy of talin with the ABS2 mutations ($rhea^{ABS2/+}$) we observed levels of talin at sites of integrin-mediated adhesion at MTJs that were slightly lower than, but near to, those observed in a wildtype embryo ($rhea^{+/+}$; Fig 4A, 4I and 4K). In comparison, in an embryo containing one copy of the $rhea^{K17E}$ allele and one copy of talin with the ABS2 mutations ($rhea^{ABS2/K17E}$) we observed loss of localization to MTJs to a similar extent as we saw in talin null embryos ($rhea^{-/-}$; Fig 4B, 4J and 4K). Furthermore, the $rhea^{K17E}$ allele, much like the talin null allele, fails to functionally complement the ABS2 mutation. In particular, embryos lacking wildtype talin but containing one copy of the $rhea^{K17E}$ allele and one copy of the ABS2 mutant talin allele ($rhea^{ABS2/K17E}$) showed phenotypes that were similar to those observed when the ABS2 mutant allele was combined with a talin null allele ($rhea^{ABS2/-}$) (Fig 3F and 3K and 3G and 3K for the ABS2 and ABS3 mutant alleles, respectively). The failure of the $rhea^{K17E}$ allele to complement or rescue flies containing the ABS2 mutation supports the interpretation that these alleles share a functional defect. In contrast, in the presence of a wildtype allele or an ABS3 mutant allele ($rhea^{ABS2/+}$ or $rhea^{ABS2/ABS3}$), a talin protein containing the ABS2 mutation can be recruited stably to sites of adhesion. Based on these data we propose that the ABS2 domain, but not the ABS3 domain, of talin plays an important role in talin recruitment to the membrane.

## Actin organization at MTJs is disrupted in talin ABS2 mutants but not talin ABS3 mutants

In late-stage fly embryos, actin is enriched at the MTJs (Fig 5A; [17]). Actin enrichment can be quantified by staining actin in muscles using phalloidin and determining the ratio of the intensity of actin staining in the cytoplasm to the actin staining intensity at the MTJs (Fig 5F). In an embryo that lacks talin, actin was not enriched at muscle ends, even in MTJs that remain attached (Fig 5B and 5F; [17]). A similar, but less severe, phenotype of loss of actin enrichment was observed in embryos that expressed only the ABS2 mutant of talin ($rhea^{ABS2/ABS2}$; Fig 5C and 5F). In comparison, in embryos that expressed only the ABS3 mutant ($rhea^{ABS3/ABS3}$) the levels of actin at muscle ends were similar to those seen in wildtype controls (Fig 5D and 5F). Similarly, in embryos that lacked wildtype talin but contained one copy of the ABS2 allele and one copy of the ABS3 mutant allele ($rhea^{ABS2/ABS3}$) the levels of actin at muscle ends were also comparable to wildtype controls (Fig 5E and 5F). Although the poor recruitment and/or maintenance of the talin protein with the ABS2 mutation complicates the interpretation of the actin phenotype of this allele, these data suggest that a talin containing the ABS3 mutation retains some ability to recruit and organize actin at sites of adhesion at muscle ends. Taken together with the near wildtype rescue of actin organization in the ABS2 and ABS3 transheterozygote mutants this data is consistent with a redundancy between the two ABSs in actin organization.

## ABS2 and ABS3 have distinct as well as overlapping roles in recruiting the integrin adhesion complex

Integrin-mediated adhesion is dependent on both the ability of integrins to establish and maintain a multiprotein complex inside the cell, and to attach to ECM molecules outside of the cell [52,53]. Talin is essential for both of these processes as it is required for the assembly of the intracellular integrin adhesion complex and also for the attachment of integrins to their ECM ligands [15,32,54,55]. To determine whether the defects observed in the ABS mutants result from disruption of either, or both, of these processes, markers for the integrin adhesion complex and the ECM were analyzed. To visualize the intracellular integrin adhesion complex in wildtype and mutant embryos two different markers were used, paxillin [56] and PINCH [57] (Fig 6A–6L). The relative localization of paxillin and PINCH to sites of adhesion at the

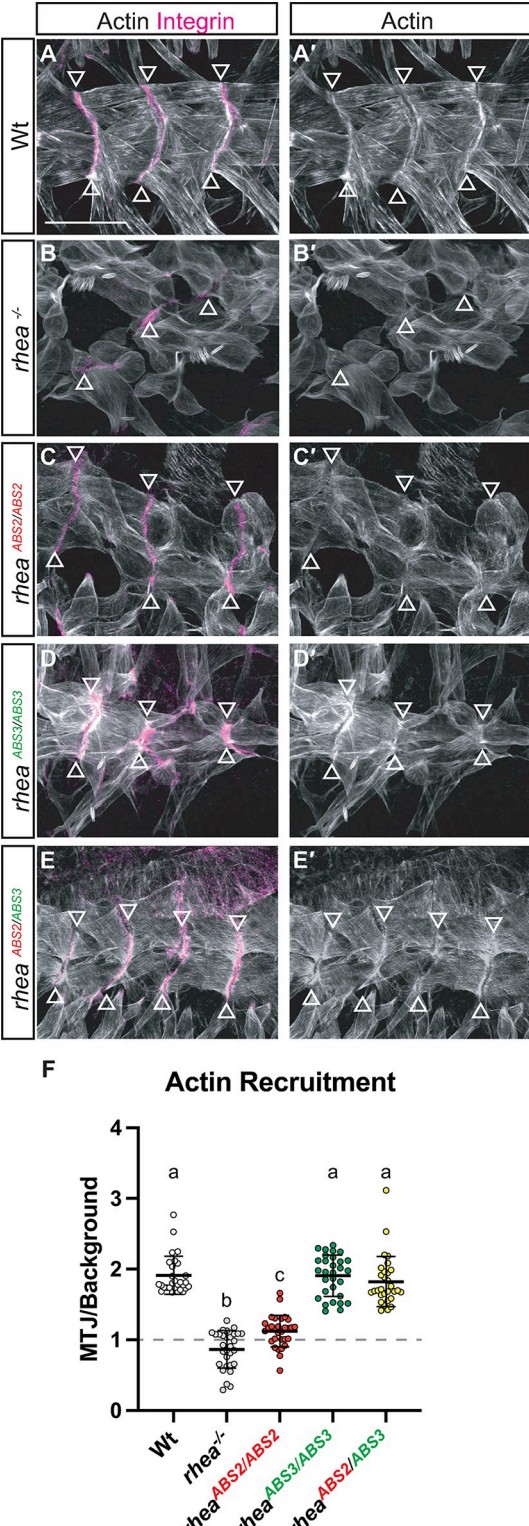

**Fig 5. Actin organization at MTJs is disrupted in ABS2 mutants but not in ABS3 mutants.** (A-E′) Confocal z-stacks of stage 17 embryos co-stained with Rhodamine-Phalloidin to label F-actin (white) and the integrin subunit αPS2 (magenta). MTJs location determined by integrin staining (arrowheads in A-E′). Scale bar: 25 μm. (F) Relative localization of F-actin to MTJs. Mean value represented by solid line, error bars show s.e.m. and dashed line indicates the absence of recruitment (n>30 per genotype). Lower-case letter (a-c) indicates a different statistical group ($P < 0.05$) using one-way ANOVA, Dunnett's post-hoc.

MTJs was quantified by determining the ratio of the intensity of paxillin or PINCH staining in the cytoplasm to their staining intensity at MTJs (Fig 6K and 6L, respectively; see materials and methods). In wildtype muscles, both paxillin and PINCH were significantly enriched at the muscle ends compared to the cytoplasm (Fig 6A and 6B; quantified in K&L respectively). This enrichment to muscle ends was not observed in embryos lacking talin, as both Paxillin and PINCH failed to concentrate at the MTJs resulting in a 1:1 ratio of cytoplasmic to MTJ staining intensity (Fig 6C and 6D; quantified in K&L respectively). A similar failure to concentrate at the MTJs was observed for either paxillin or PINCH in embryos that lacked endogenous talin but expressed the ABS2 mutant ($rhea^{ABS2/ABS2}$; Fig 6E and 6F; quantified in 6K and 6L, respectively). However, near wildtype levels of enrichment at the MTJs were seen in embryos that lacked endogenous talin but expressed the ABS3 mutant ($rhea^{ABS3/ABS3}$; Fig 6G and 6H; quantified in 6K and 6L respectively). Notably, in embryos that lacked wildtype talin but contained one copy of the ABS2 and ABS3 mutant alleles ($rhea^{ABS2/ABS3}$) paxillin levels, but not PINCH levels, at the MTJs were partially reduced compared to wildtype control ($rhea^{ABS2/ABS3}$; Fig 6I and 6J; quantified in 6K and 6L respectively). These results suggest that Paxillin recruitment to sites of integrin-mediated adhesion is dependent on ABS2 function to a greater extent than PINCH recruitment, while ABS3 is not required for the recruitment of either paxillin or PINCH.

To test whether the talin ABSs were required for integrin-mediated attachment to the ECM we visualized the staining of the ECM ligand Tiggrin at muscle ends. In wildtype MTJs, Tiggrin and αPS2-integrin localization at MTJs overlapped (Fig 6M; visualized using a staining intensity profile in P). It has been previously shown that mutations that disrupt integrin-mediated attachment to the ECM resulted in clear separation of Tiggrin from the integrin at MTJs [42, 48]. However, in embryos that lacked endogenous talin but expressed either the ABS2 or ABS3 mutant talin, Tiggrin and αPS2-integrin localization at MTJs still overlapped (Fig 6N and 6O, respectively; visualized using a staining intensity profile in P). These results suggest that neither ABS2 or ABS3 are required for establishing or maintaining the linkage of integrins to their ECM ligands.

## Vinculin overexpression ameliorates the phenotype of talin ABS3 domain mutants

One of the main proposed functions of the talin ABSs is to facilitate force transmission across the protein by linking to actin. Force transmission across the talin protein is thought to lead to stereotypic unfolding of the repeating helical bundles that constitute the large rod domain of talin, a conformational change that exposes cryptic protein-protein binding sites [23,27–29,31,58]. In particular, talin contains a large number of binding sites in its rod domain for the scaffolding protein vinculin [19,23,59,60] that are not available in the folded conformation but are available in the unfolded conformation [27,29,61,62]. It has been proposed that the exposure of these cryptic binding sites in talin and the subsequent binding of vinculin to talin plays an important role in the consolidation and maturation of integrin-based adhesions [30,31,62]. According to this proposed model, the binding of actin through ABS3 allows actin to pull on talin and expose cryptic vinculin binding sites that, when bound by vinculin, promote actin binding by ABS2 and adhesion stabilization. Key evidence in support of this model was provided by experiments showing that expression of an activated vinculin can compensate for loss of actin binding through ABS3 [23]. To reproduce this experiment in flies we drove ectopic vinculin expression in our ABS2 and ABS3 mutant alleles. We are not able to use activated vinculin since it has been shown that, in flies, its expression has dominant negative effects resulting in loss of function phenotypes in integrin-mediated adhesion [63]. Instead, we employed

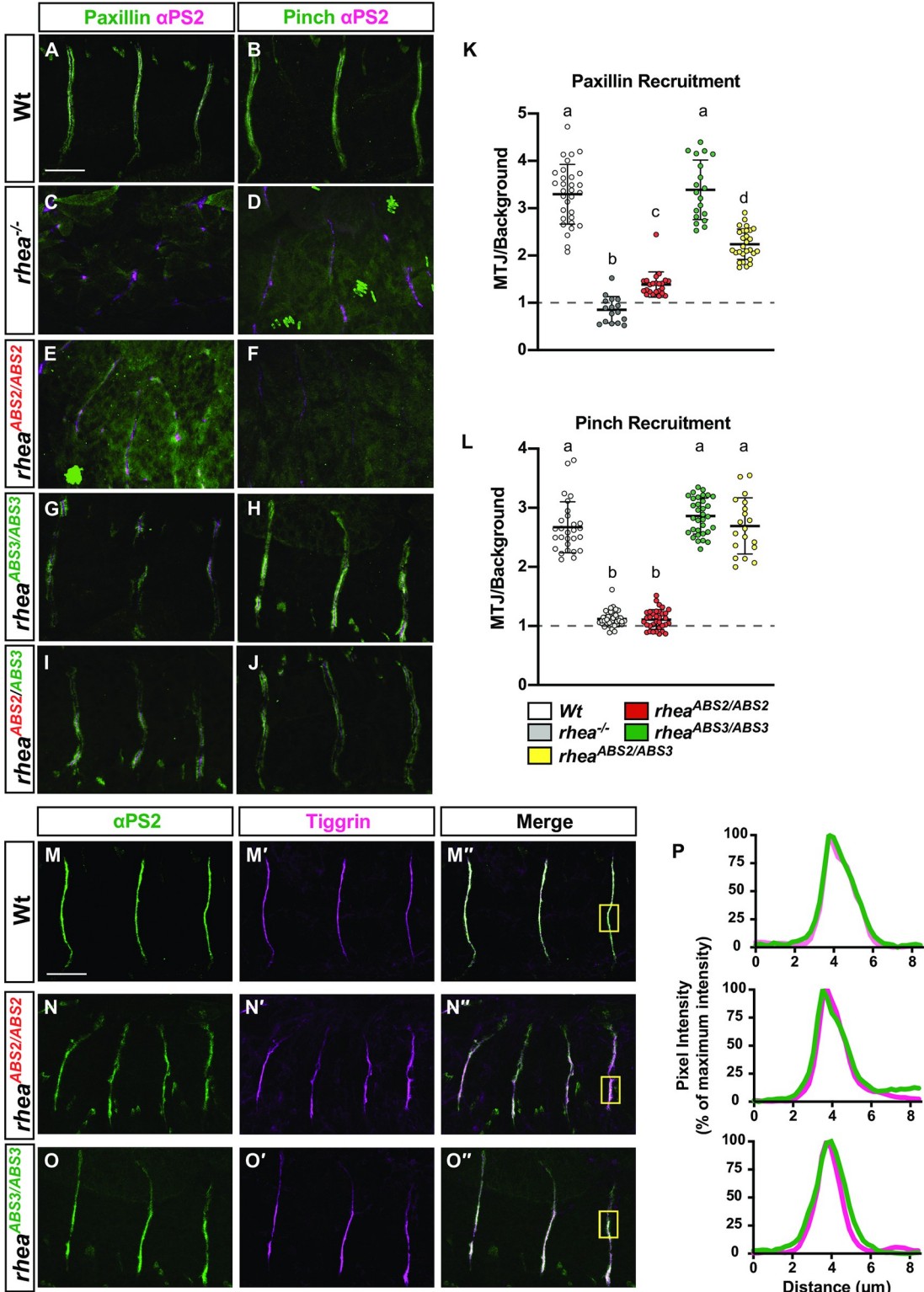

**Fig 6. ABS2 and ABS3 roles in recruiting the integrin adhesion complex and attachment to the ECM ligand Tiggrin.** (A–J) Confocal z-stacks of MTJs in stage 17 embryos stained for IAC components paxillin and PINCH (green, A-J), the integrin subunit αPS2 (magenta, A-J and green, M—O″). (K&L) Relative localization of Paxillin (K) and PINCH (L) to MTJs localized by αPS2 staining. Mean value represented by solid line, error bars show s.e.m. and dashed line indicates the absence of recruitment (n>15 per genotype). Lower-case letter (a-d) indicates a different statistical group (*P*<0.05) using one-way ANOVA, Dunnett's

post-hoc. (M-O″) Confocal z-stacks of MTJs in stage 17 embryos stained for the ECM ligand tiggrin (magenta, M′-O″). (P) Pixel intensity profiles of tiggrin (top panel Wt, middle panel *rhea*^ABS2/ABS2^ and lower panel *rhea*^ABS3/ABS3^) across the widths of the boxed areas indicated in the corresponding images. Pearson correlation coefficient ($r$) > 0.7 for all genotypes. Scale bar: 25 μm.

overexpression of wildtype vinculin in muscles of zygotic talin mutants, that contain maternal talin and therefore have weaker phenotypes than the complete loss of talin, using vinculin overexpression in wildtype flies as the control. Previous work characterizing vinculin overexpression suggested that expression of constitutively active vinculin, but not overexpression of wildtype vinculin, had a severe impact on viability (63). In contrast, we noted that overexpression of wildtype vinculin also resulted in a decrease in viability of about 30% across all genotypes (Fig 7B). Nonetheless, we were able to assess the impact of overexpression of vinculin in the background of the different talin mutants. In combination with the talin null allele, or an ABS2 mutant allele (*rhea*^-/-^ or *rhea*^ABS2/ABS2^) we noted had no impact on the phenotype (Fig 7A–7C). In comparison, overexpression of vinculin in the background of a talin ABS3 mutant allele (*rhea*^ABS3/ABS3^) substantially ameliorated the phenotype as some flies were able to survive longer, and equalized larval mobility between ABS3 mutants and control wildtype larvae (Fig 7A–7C). These results suggest that overexpression of vinculin, despite having a slight negative impact on viability, can nevertheless ameliorate the phenotype caused by disruption of the ABS3 domain of talin.

We also noted in these experiments that the phenotype observed in the ABS3 mutant background where vinculin is overexpressed was, in terms of mobility, and viability, indistinguishable from what was observed in ABS2/ABS3 transheterozygotes mutants (Fig 7A–7C). In addition, overexpression of vinculin did not modify the phenotype of ABS2/ABS3 transheterozygotes mutants (Fig 7A–7C). These data implied that the synergistic interaction between the ABS2 and ABS3 mutant talin proteins could be mimicked by overexpressing vinculin. Further support for this idea came from experiments studying vinculin recruitment in talin mutants. Vinculin recruitment was analyzed using two different approaches: by ectopic expression of a RFP tagged version of Vinculin that was used in the overexpression experiments (Vinculin-TagRFP; (63)) or by analyzing the distribution of a genomically GFP-tagged (Vinculin::GFP; [64]), allele of vinculin Both approaches showed that in a talin null background vinculin recruitment to sites of adhesion at muscle attachment sites was severely curtailed compared to wildtype controls (Fig 7D, 7E and 7J, 7K, quantified in 7I). This defect in vinculin recruitment was nearly as severe in talin ABS2 mutants (Fig 7F and 7L, quantified in 7I). However, in talin ABS3 mutants or in transheterozygote mutants vinculin recruitment was substantially better compared to the null or ABS2 mutants (Fig 7G, 7H and 7M, 7N, quantified in 7I). The observation that ABS3 mutants retained some capacity to support vinculin recruitment to sites of adhesion helps explain why they are amenable to rescue by vinculin overexpression. This result suggests that the general relationship between talin ABS3 mutants and vinculin activity is conserved between mammals and flies.

## In vivo evidence for the functional importance of talin dimerization

Talin has long been thought to form a dimer [24,65–68] but this has not been demonstrated in a functional capacity *in vivo* before. The substantial rescue, in terms of both talin recruitment and developmental phenotypes, that was observed by combining the talin ABS2 and ABS3 mutants suggested the existence of a talin dimer that contained one ABS2 and one ABS3 protein. Specifically, we hypothesized that a talin dimer composed of one ABS2 and one ABS3 mutant talin proteins could perform more functions than a talin dimer composed of two ABS2 or two ABS3 mutant talin proteins. For example, we hypothesized that while a talin dimer

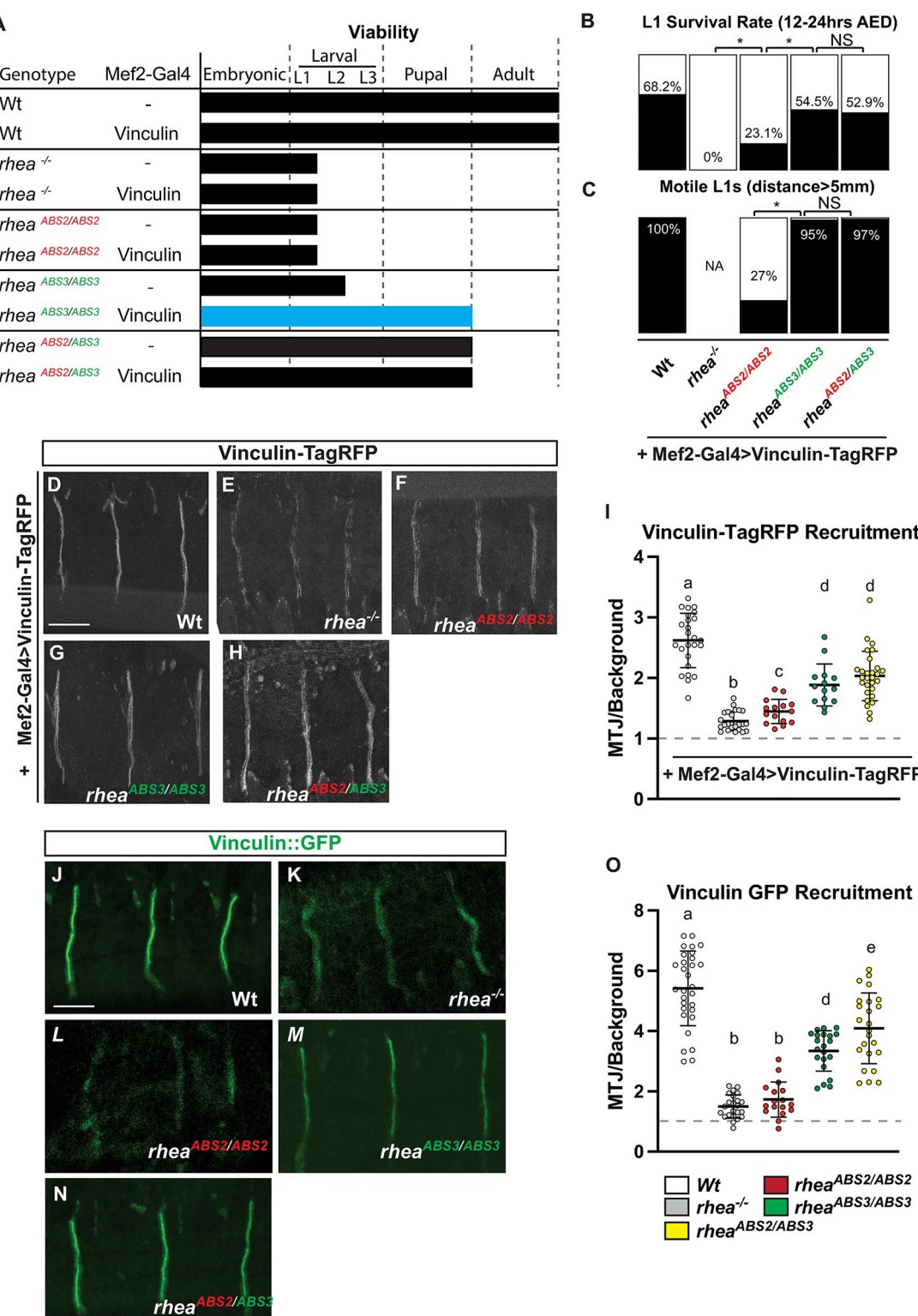

**Fig 7. Vinculin overexpression partially rescues talin ABS3 mutants' viability phenotype.** (A) Viability of Mef2::GAL4 flies with or without a UAS promoter expressing vinculin-TagRFP in genetic backgrounds as shown. (B) L1 Survival rate. Percentage of L1 larvae successfully hatching 12–24 hours after egg deposition (AED). (C) L1 motility. Percentage of L1 larvae capable of moving at least 5 mm away from hatching site. (D-H) Muscles in live wildtype stage 17 embryos expressing the TagRFP -tagged vinculin (scale bar: 25 μm). (I) Relative localization of vinculin- TagRFP to MTJs in live wildtype stage 17 embryos expressing the

TagRFP -tagged vinculin. (J-N) Native vinculin expression in stage 17 embryos using genomically GFP-tagged vinculin (Vinculin::GFP). (O) Relative localization of Vinculin;;GFP to MTJs in live wildtype stage 17 embryos expressing. Mean value represented by solid line, error bars show s.e.m. and dashed line indicates the absence of recruitment (n>15 per genotype). Lower-case letter (a-c) indicates a different statistical group (P<0.05) using one-way ANOVA, Dunnett's post-hoc.

composed of two mutant ABS2 proteins was not recruited to sites of adhesion and was unable to function in adhesion, a dimer composed of one ABS2 and one ABS3 mutant was recruited to, and was partially functional at, sites of adhesion (Fig 4K). To directly test this hypothesis, we utilized the previously described R2531G talin mutant, that is known to block talin dimerization [17]. In addition to blocking dimerization the R2531G mutation disrupts the function of the ABS3 domain [17,24]. This allowed us to compare the impact of a simple disruption in ABS3 versus disruption of ABS3 coupled with an inability to form dimers. For this experiment we relied on two transgenes we previously generated and characterized that expressed, under the control of ubiquitous promoter, a talin rescue transgene containing either the ABS3 or the R2531G mutations ([17]; Fig 8A–8C; Quantified in J). As shown in our previous study, while introducing a wildtype rescue transgene under the control of a ubiquitous promoter fully rescued embryonic muscle detachment of talin null embryos, introducing either the mutant ABS3 or the R2531G transgenes provided only a partial rescue of muscle detachment in the talin null background ([17]; Fig 8D–8F; Quantified in J). Furthermore, when we introduced the mutant ABS3 transgene into the background of the ABS2 mutant we significantly rescued the muscle attachment and recapitulated the transheterozygote phenotype (Fig 8H; quantified in 8J). These results validate our approach and confirm the specificity of the synergistic interaction between ABS2 and ABS3 talin mutants. In contrast, when the R2531G transgenes were introduced into the background of ABS2 mutant embryos we did not observe the strong rescue that was seen with the ABS3 transgene but a mild rescue similar to the effect of introducing the R2531G transgene into talin null flies (Fig 8F and 8I; quantified in 8J). These results suggest that the ability of talin to dimerize is key for the synergistic interaction we observe between talin ABS2 and talin ABS3 mutant alleles in line with the idea that talin functions as a dimer (Fig 8K).

## Discussion

A central defining feature of integrin-based adhesions is their ability to bind to and organize the cytoskeleton, an ability that is key to their function in mediating cell adhesion, supporting tissue architecture, promoting cell motility, and enabling mechanosensing [52,53,69,70]. A major challenge in understanding how integrins link to cytoskeletal proteins, in particular actin, is that there are many possible ways to establish such a link that involve different cytoskeletal linker proteins that are part of the integrin adhesion complex. There are multiple possible explanations for why there are so many possible ways for the integrin adhesion complex to link to the cytoskeleton including: increased robustness due to redundancy, introducing the possibility of functional specialization, providing additional regulatory nodes that can be used to fine tune adhesions. Here we have attempted to explore these possibilities, taking advantage of the presence of multiple, well characterized, distinct binding sites for actin in talin. Our genetic approach indeed argues that having multiple actin-binding sites provides a degree of robustness and allows both functional redundancy and specialization.

We previously used a different, transgene-based rather than a CRISPR-based, rescue strategy to study the role of ABS3, this analysis provided us with initial indications of the possibility that there is some functional specialization of the talin actin binding sites [17]. The development of CRISPR based technology has not only allowed us to characterize, for the first time,

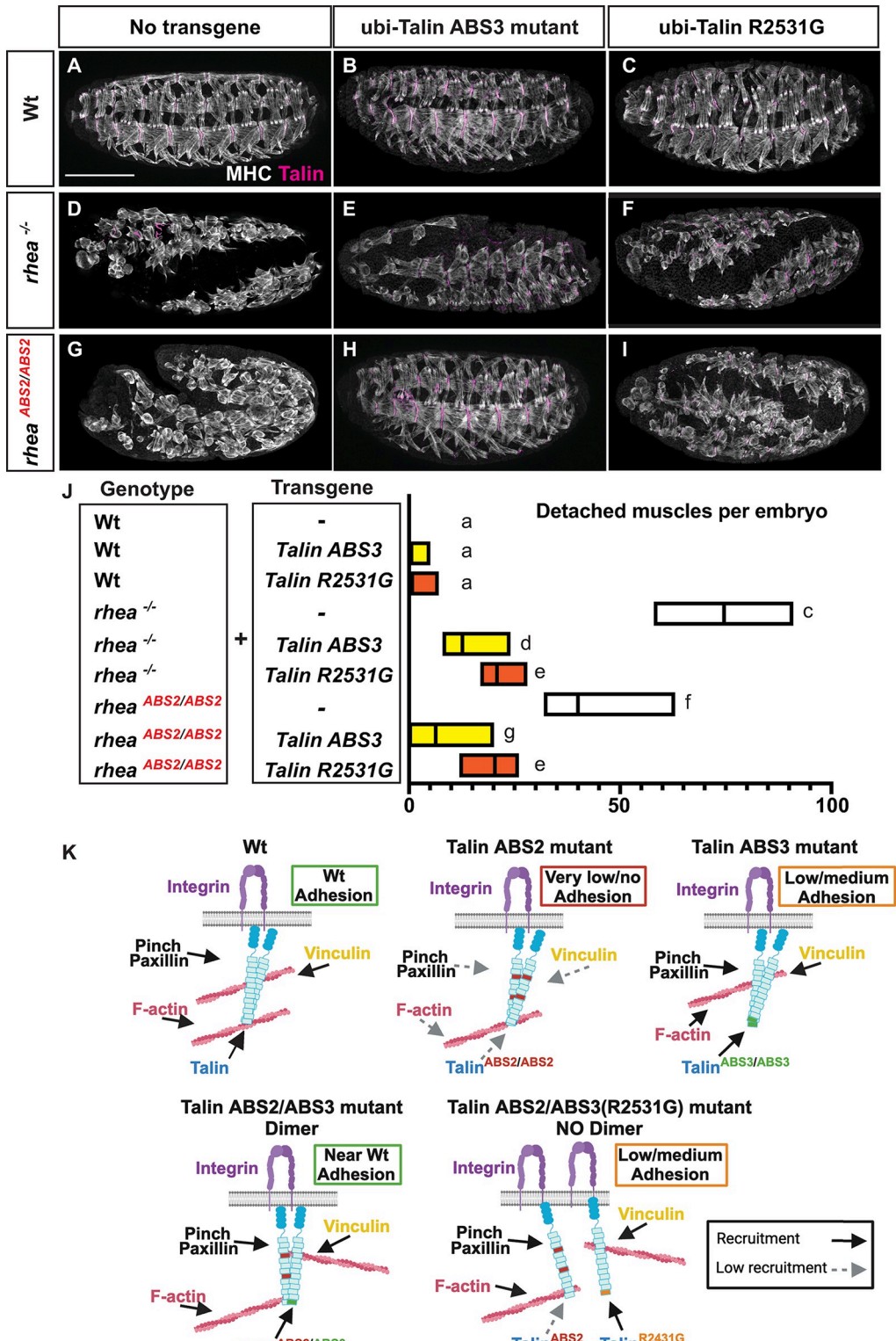

**Fig 8. ABS2/ABS3 transheterozygote interaction requires talin dimerization.** (A-I) Confocal images of stage 17 whole-mount embryos stained with muscle cytoskeleton marker MHC (white) and Talin (magenta) ubiquitously (ubi) expressing: no transgenes (A,D,G), Talin ABS3 mutant protein (B,E,H) and Talin R2531G protein (C,F,I) in Wt (*rhea*[+/+], A-C), *rhea* null (*rhea*[-/-], D-F) and ABS2 mutant (*rhea*[ABS2/ABS2], G-I) backgrounds. Scale bar: 50 μm. (J) Number of detached muscles per embryo. Floating bar chart representing minimum and maximum value per genotype (n>10).

Mean value represented by a solid line. Lower-case letter (a-e) indicates a different statistical group ($P<0.05$) using one-way ANOVA, Dunnett's post-hoc. (K) Schematic model, created with BioRender.com, illustrating the role of ABS2 and 3 in integrin mediated adhesion. Solid arrows indicate recruitment to the MTJs whereas dashed ones represent the lack of recruitment.

the *in vivo* role of the ABS2 domain of talin but also to greatly extend this earlier analysis of ABS3 and carry out a direct functional comparison of ABS2 and ABS3. Moreover, the use of CRISPR generated alleles has made it possible to explore genetic interactions between mutations in ABS2 and ABS3, which proved very informative. In particular, genetic analysis of the transheterozygotic mutant combination, containing one copy of each mutant allele, revealed a substantial degree of intragenic complementation between mutations in ABS2 and ABS3. This finding directly illustrated both the substantial level of functional overlap between the ABS domains, represented by the degree to which each ABS mutant allele was able to compensate for the other, but also revealed the limits of this redundancy, represented by the defects seen in the transheterozygotes that lead to eventual lethality during late larval stages.

Another important use for the ABS2/ABS3 transheterozygote was in providing a means to study the function of the ABS2 mutant protein in adhesions despite the inability of this mutant, when it was the only version of talin made in the animal, to localize to, or be retained at, adhesions. This was useful since the inability of the talin ABS2 mutant protein to localize to adhesions acted to mask possible remaining functionality of this allele. Our genetic evidence suggests that the ABS2 mutant, in the presence of a talin protein that is able to localize to the membrane (such as the ABS3 mutant), can be found at adhesions and function in certain capacities. Collectively, our evidence identifies three specific roles for the talin ABS2 domain that are either partially or fully non-functionally redundant with the ABS3 domain, the recruitment and/or stabilization of talin at the membrane, the recruitment and/or stabilization of paxillin at the membrane, and the establishment and/or maintenance of actin organization at MTJs. While we did not identify roles for the ABS3 domain that were non-redundant with the ABS2 domain we found that mutations in ABS3 but not in ABS2 could be rescued by vinculin overexpression, consistent with previous cell culture data [23] where the role of ABS3 in focal adhesion assembly could be bypassed by the expression of active vinculin. Based on this cell culture data it was proposed that a key role of ABS3 was to drive the initial unfolding of the talin rod to promote vinculin binding by talin. Although our data is in line with this idea it is known that vinculin is not required for integrin-based adhesion in flies, but that it serves a redundant but powerful regulatory role when active, specifically in promoting adhesion formation [63]. It was further proposed that, in *Drosophila*, vinculin is able to bind to talin even in the absence of the sort of mechanical force that stretches talin to reveal cryptic binding sites [63]. Taken together with these previous observations, our results support the idea that ABS3 promotes adhesion by allowing the unfolding of talin in response to mechanical force, which facilitates the exposure of cryptic protein-binding sites. However, providing an excess of vinculin activity can act to bypass the need for mechanical stretching of talin, allowing for some compensation or rescue for the absence of ABS3 based force-mediated unfolding of talin.

Talin was initially speculated to form antiparallel dimers based on computer assisted structural modeling [71]. Subsequent studies using structural and biophysical analysis of platelet derived native talin provided direct evidence that, at least in platelets, the majority of talin protein existed in the form of antiparallel dimers [65]. Subsequent work identified and characterized a C-terminal region in talin that mediated its dimerization [17,24,66,72]. However, since disrupting the talin dimerization domain also abrogates the actin binding ability of the ABS3 domain it has been difficult to provide direct evidence that talin acts as a dimer *in vivo*. Here we have taken advantage of the functional complementation of ABS2 and ABS3 mutants as

well as an integrin dimerization mutant that we previously made and characterized to provide direct evidence that talin functions as a dimer *in vivo*. Importantly, our results imply that each talin protein within a dimer has the ability to function with a degree of independence from the other talin protein. This is illustrated by the observation that although each of the ABS2 and ABS3 mutants in the transheterozygote background have compromised function, they act in a synergistic manner in the context of a dimer and exhibit substantially more functionality than either mutant by itself.

Our work shows that, *in vivo*, both ABS2 and ABS3 are essential for integrin-based adhesion and viability. Loss of the talin ABS2 results in a more severe phenotype presumably due to it having a role in the ability of talin to localize to, or be stabilized at, integrin-based adhesions (Fig 8K). According to current models, the initial engagement of talin with actin, which allows the molecule to stretch and reveal cryptic protein interaction motifs, occurs through ABS3 [23,30] which might predict a stronger phenotype for a mutation impinging on ABS3. There are two possible explanations for this lack of a stronger phenotype, one is the existence of residual actin binding and thus activity with the KVK/DDD mutation in ABS3 that we utilized in this study. In particular, although the KVK mutation is widely used in cell culture studies and is the strongest known mutation in this domain that does not interfere with other functions of talin, evidence suggests it has residual ability to bind actin [17,24]. Importantly, arguing against this explanation is our previous finding that an allele of talin that completely deletes the ABS3 domain (*rhea*$^{15\text{-}39}$; [17]) shows similar hypomorphic phenotypes to those we observed when we characterized the KVK/DDD mutation. A second explanation for the absence of a stronger phenotype with ABS3 comes from our work and earlier findings [23] that suggest that there are alternative mechanisms that can compensate for loss of ABS3 based actin binding, such as vinculin activation. Moreover, even though in this sequence of events ABS2-based actin binding occurs after ABS3-based actin binding, ABS2 plays a more essential role, in the sense that it cannot be compensated for by alternative mechanisms. Consequently, while loss of ABS3 gives rise to an intermediate phenotype, loss of ABS2 leads to a null-like phenotype. Notably, it was previously proposed [30] that ABS3 function might be largely regulatory, as it controls the activation of ABS2 so as to prevent its premature engagement with actin. In addition, recent findings provided a mechanism for how the ABS3 domain of talin can perform such a regulatory role. Specifically, ABS3 binds to actin poorly until subjected to the type of directional mechanical forces generated by the cytoskeleton which induces strong actin binding [73]. This ability of ABS3 to form "catch-bonds" with actin was proposed to be an important way of controlling adhesion growth and actin organization. Based on our findings we propose that as adhesion complexes contain multiple ways to connect to the cytoskeleton, this offers the ability for functional, spatial, regulatory, and temporal specialization for specific cytoskeletal binding events. In addition, the ability of separate cytoskeletal linkage mechanisms to act synergistically adds an intriguing new level of functional complexity to cell adhesion.

## Materials and methods

### Molecular biology

The generation of the ABS2 and ABS3 mutants described here (Fig 1) are based on a modified version of the following protocols: (http://flycrispr.molbio.wisc.edu, [47]). For ABS2 the following target sequences were respectively used for S1 and S2: 5′-GTAAGGCAGCGGAG-GAACTTCGG-3′ 5′-GGCTACTTCGGACTTGGTGCAGG-3′. The 5' and 3' donor arms (Fig 1D) were synthesized by Integrate DNA Technologies (IDT) and assembled by Gibson cloning in the pHD-ScarlessDsRed plasmid (gift from Kate O'Connor-Giles, Addgene plasmid # 64703; http://n2t.net/addgene:64703; RRID:Addgene_64703). pHD-ScarlessDsRed is a donor

vector with a 3xP3-DsRed marker cassette flanked by PBac transposon ends allowing for easy identification of engineered lines. For ABS3 the following target sequence was used 5′-GCTTATTTCCACAGCCAAGCAGG-3′. The single-stranded oligonucleotides (ssODNs) was generated by IDT. All targeting gRNAs were cloned by annealing the corresponding target sequence oligonucleotides into the pU6-BbsI-chiRNA plasmid[47] via the BbsI restriction sites. For both mutants the CRISPR injection mix containing the double-stranded donor DNA (500 ng/μl) along with targeting plasmids (100 ng/μl) was sent to Bestgene Inc. for injection. ABS2 mutant transgenic flies were identified by eye color screening using the DsRed gene included in the donor DNA. This visible eye marker was then excised by crossing ABS2 mutants to P{Tub-PBac\T}2 flies expressing piggyBac transposase leaving behind a residual TATA sequence. As the selection cassette was inserted into a native TATA site that was deleted in the donor vector the cassette was seamlessly removed. ABS3 mutant transgenic flies were identified by crossing P0 flies (n = 10) to third chromosome balancers then letting F1 flies interbreed. Resulting F2 progeny were crossed to *rhea* null flies for complementation test (100 single F2 flies crosses per P0 line). F2 candidates unable to complement *rhea* null were then sequenced for the presence of the ABS3 mutation. The resulting mutants were then recombined in an FRT2A background and sequenced using standard techniques (Fig 1F and 1G).

For the quantitative real-time PCR (qPCR) total RNA was isolated from whole flies using TRIzol. A total of 0.5 μg total RNA was converted into cDNA using the qScript cDNA synthesis kit (Quanta Biosciences). Subsequently, qPCR was performed using iQ SYBR Green Supermix (BIO-RAD). Talin mRNA levels were averaged between three independent experiments performed four times and normalized to β-tubulin expression. Primers used for Talin were located 3′ to the K17E mutation and were as follow 5′-GCCAGAACAATACTTTGGGTCG-3′ and 5′-AACTGGGCATTTCGCTGGAA-3′. β-tubulin expression was determined using the primer pair 5′-ATCATCACACACGGACAGGA-3′ and 5′-GAGCTGGATGATGGGGAGTA-3′.

## Fly stocks and genetics

Except for viability/survival, motility and vinculin recruitment all experiments were performed in *rhea* mutant background such that wild-type maternal contributions of Talin were eliminated using the following alleles: *rhea79a* (null allele, [15], *rheaABS2*, *rheaABS3* or *rheaK17E* [43] in conjunction with the dominant female sterile technique [74]. Females of the genotype yw, hs-Flp/+;; *rhea* mutant, FRT2A/OvoD1, FRT2A were subjected to a heatshock regime during the larval stages to generate mosaic animals in order to give rise to *rhea* mutant oocytes. Mosaic virgins were assessed for wing blisters and then crossed to the corresponding *rhea* mutant/ TM6b, dfd-GMR-nvYFP males. Embryos without the fluorescent balancer were selected for analyses. ABS2 and 3 transheterozygotes mutants (*rheaABS2/ABS3*) were generated by crossing heatshocked yw, hs-Flp/+;; *rheaABS2*, FRT2A/OvoD1, FRT2A virgins to *rheaABS3*/ TM6b, dfd-GMR-nvYFP males. Full-length Vinculin with a C-terminal TagRFP tag [63] were driven in the muscles using the mef2-GAL4 driver. Vinculin::GFP allele was a gift from Yohanns Bellaïche [64]. GFP-tagged full-length talin construct (Talin-GFP, [75]) as well as Talin ABS3 (Talin KVK-DDD in [17]) and Talin R2531G [17] where ubiquitously expressed via the Ubiquitin-E63 gene promoter [75]. The ubi::TalinGFP line [47] used in rescue experiments has been shown to phenotypically rescue null *rhea* alleles fully to the end of larval stages but for currently unknown reasons fails to rescue *rhea* null mutants to viability.

## Viability and motility assay

5 mm black dots were drawn at the back of 60x15 mm 2% agar petri dishes. 12–24 hours old eggs were then placed on each dot and monitored for 5 days. Resulting larvae were assessed for

viability: most advanced developmental stage reached by at least ten larvae, L1 survival rate: percent eggs of a given phenotype hatching and L1 motility: percent L1s moving outside of 5 mm black dot after 1 hour minimum.

## Confocal immunofluorescence imaging and image analysis

For all antibody stainings embryos were collected for 24 h at 25°C and heat fixed [48]. For rhodamine phalloidin staining, embryos were fixed in 4% formaldehyde followed by devitellinization in 80% ethanol. Antibody stainings were carried out according to standard procedures. Antibodies used were against the following proteins: MHC (mouse, 1:200; from Dan Kiehart, Duke University, Durham, NC), PINCH (rabbit, 1:1000; from Mary Beckerle, Huntsman Cancer Institute, UT), Talin (rabbit, 1:500), αPS2 (rat, 1:100; 7A10), Paxillin (rabbit, 1:1000) (Yagi et al., 2001)[56], tiggrin (mouse, 1:500; from Liselotte Fessler, UCLA, CA) and GFP (rabbit, 1:1000; A6455, Invitrogen). Fluorescently conjugated Alexa Fluor 488, Cy3 and Cy5 secondary antibodies were used at 1:400 dilution (Molecular Probes). Images were collected using an Olympus FV1000 inverted confocal microscope. For all micrographs of whole embryos, larvae or of MTJs, z-stacks were assembled from 8–12 0.5 μm confocal sections. Statistically significant differences were assessed by two-tailed Student's t-tests in all cases, except when we sought to compare between multiple constructs, where one-way ANOVA was used. Differences between 2 percentage groups was assessed by chi-square test. Statistical analysis was carried out using Prism4 software (GraphPad, La Jolla, CA). Relative protein localization per larva was determined by the average pixel intensity of 3 MTJs divided by the average pixel intensity of the corresponding background area: zone of equal size directly anterior to the MTJ (Raw data shown in S1 Data). For intensity traces across MTJs (Fig 6P), the ImageJ plot profile tool was used to determine the average signal intensity across the boxed area indicated on the images. Intensity curves were obtained from unprocessed greyscale images so that first the peak intensity of each channel across the area of interest was set as 100%. Each curve was then normalized to the average intensity measured outside of the MTJ. Aggregates and muscle defects were counted manually using ImageJ.

## Muscle detachment analysis

For embryonic muscle detachment analysis, z-stack projections of whole mount stage 17 embryos of each genotype stained with α-MHC were selected. Total number of muscles that were clearly detached and had a round shape was scored manually using ImageJ cell counter function (Raw data shown in S1 Data). For larval muscle detachment analysis desired genotypes were crossed into a fluorescent Z-disc marker (α-actinin-GFP). Resulting larvae were then heat-fixed by submerging them into 65°C PBS for 1–2 s and mounted in halo carbon before immediately being imaged [76].

## Recombinant protein expression and purification

For biochemical analysis, pET151 plasmids for fly talin R4 Wt (residues 932–1062), R4-3E (residues 932–1062, K941E/Q942E/Q949E R7R8), R7R8 Wt (residues 1367–1666), R7R8-3E (residues 1367–1666 K1510E/R1520E/K1532E), R13 Wt (residues 2304–2489) and R13-3D (residues 2304–2489 K2450D/V2451D/K2452D) were produced as codon optimized synthetic genes by GeneArt.

15N-labeled proteins were prepared by transforming talin constructs into BL21(DE3)* E. coli cells and grown in 10 mL of LB + ampicillin at 37°C overnight as starter cultures and then in 2M9 minimal medium containing 15N-ammonium chloride at 37°C until an OD600 of 0.7–0.8. The cells were induced with 1 mM IPTG, then the cultures were incubated overnight

at 20˚C. The following day, harvested cells pellets were resuspended in 20 mM Tris-HCl, pH 8, 500 mM NaCl, and 20 mM imidazole, and lysed by sonication. Protein purification was achieved by nickel affinity chromatography using a 5 ml HisTrap HP column (GE Healthcare). The eluted proteins were dialyzed into 20 mM Tris-HCl, pH 8, 50 mM NaCl with AcTEV protease (Invitrogen) to remove the His-tag. After overnight dialysis, the proteins were purified with a HiTrap Q HP cation (R7R8 and R13) or HiTrap S HP anion (R4) exchange column (GE Healthcare).

### Circular Dichroism (CD) experiments

Protein samples at a concentration of 50 μm were placed in a quartz cuvette and far-UV spectra were collected between 260 and 200 nm wavelengths using a JASCO J-715 spectropolarimeter at 20 and 95˚C.

### NMR experiments

15N-labeled protein samples at 0.15 mM were placed in Shigemi NMR tubes containing 10% D2O and 2D 1H-15N HSQC spectra were collected at 298 K on Bruker Avance III 600 MHz NMR spectrometer equipped with CryoProbe.

## Supporting information

**S1 Fig. Biochemical characterization of the fly Talin R4, R8 and R13 mutants.** (A-C) 1H,15N HSQC spectra recorded at 25˚C of the Wt (black) and mutant (red) variants of (A) R4, (B) R7-R8 and (C) R13 domains. (D-F) CD spectra of the Wt (black) and mutant (red) variants of (D) R4, (E) R7-R8 and (F) R13 domains at 20˚C and 90˚C.
(TIF)

**S1 Data. Raw data of all muscle detachment and relative protein localization experiments.**
(XLSX)

## Acknowledgments

We thank Yohanns Bellaïche, Rodrigo Fernandez Gonzalez and the Bloomington Drosophila Stock Center (NIH P40OD018537) for providing us with numerous fly lines. We thank Pierre-Yves Musso for helpful discussions and reviewing all the figures.

## Author Contributions

**Conceptualization:** Darius Camp, Veronika Solianova, Benjamin T. Goult, Guy Tanentzapf.

**Formal analysis:** Darius Camp, Bhavya Venkatesh, Veronika Solianova, Lorena Varela, Benjamin T. Goult.

**Funding acquisition:** Benjamin T. Goult, Guy Tanentzapf.

**Investigation:** Darius Camp, Bhavya Venkatesh, Veronika Solianova, Lorena Varela, Benjamin T. Goult.

**Methodology:** Darius Camp, Veronika Solianova, Lorena Varela, Benjamin T. Goult, Guy Tanentzapf.

**Project administration:** Darius Camp, Benjamin T. Goult, Guy Tanentzapf.

**Resources:** Darius Camp, Veronika Solianova, Lorena Varela, Benjamin T. Goult, Guy Tanentzapf.

**Supervision:** Darius Camp, Benjamin T. Goult, Guy Tanentzapf.

**Validation:** Darius Camp, Bhavya Venkatesh, Benjamin T. Goult, Guy Tanentzapf.

**Visualization:** Darius Camp, Veronika Solianova, Benjamin T. Goult.

**Writing – original draft:** Guy Tanentzapf.

**Writing – review & editing:** Darius Camp, Bhavya Venkatesh, Lorena Varela, Benjamin T. Goult, Guy Tanentzapf.

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
