## [Decision Letter · Decision Letter 0]

26 Oct 2023

Dear Dr Tanentzapf,

Thank you very much for submitting your Research Article entitled 'The actin binding sites of talin have both distinct and complementary roles in cell-ECM adhesion' to PLOS Genetics.

The manuscript was fully evaluated at the editorial level and by independent peer reviewers. The reviewers appreciated the attention to an important problem, but raised some substantial concerns about the current manuscript. Based on the reviews, we will not be able to accept this version of the manuscript, but we would be willing to review a much-revised version. We cannot, of course, promise publication at that time.

If you decide to revise the manuscript for further consideration at PLOS Genetics, please aim to resubmit within the next 60 days, unless it will take extra time to address the concerns of the reviewers, in which case we would appreciate an expected resubmission date by email to plosgenetics@plos.org.

We are sorry that we cannot be more positive about your manuscript at this stage. Please do not hesitate to contact us if you have any concerns or questions.

Yours sincerely,

Pablo Wappner

Academic Editor

PLOS Genetics

Gregory P. Copenhaver

Editor-in-Chief

PLOS Genetics

Reviewer's Responses to Questions

**Comments to the Authors:**

Reviewer #1: In this manuscript, Camp et al have conducted a series of experiments to study the function of two actin binding sites on Talin using Drosophila in vivo. While the function of different domains in Talin have been studied through in vitro experiments or via overexpression-based experiments in vivo, this is the first time extensive structure function analyses are being conducted by manipulating the endogenous gene in a model organism in vivo, to my knowledge. The authors took advantage of the scarless gene editing technique in Drosophila to introduce a series of mutations that inhibit the function of two previously identified actin binding sites in fly Talin (encoded by the rhea gene). The authors performed a series of control experiments to ensure that these mutations do not affect the global structure of Talin by NMR and also demonstrated that the mutant lines they established are clean alleles via rescue and qPCR experiments. Based on a series of experiments, which I think have been conducted with rigor, the authors conclude that the ABS2 and ABS3 domains of Talin have complementary but distinct/non-redundant functions. Specifically, the ‘intragenic complementation’ phenomenon that the authors saw when the two mutations were tested in trans is of great interest, providing strong evidence that ABS2 and ABS3 can act somewhat independent from each other in a single molecule, and that the dimerization ability of Talin allows a molecule that is defective in one of these domain to dimerize with another molecule that is defective in the other domain to still retain significant function as a protein complex. The careful comparison of subcellular localization analysis of components of the integrin-adhesion complex and genetic interaction they observed between Talin mutants and Vinculin adds additional mechanistic insights and values to this paper. The use of a number of complementary genetic techniques (e.g. zygotic mutant analysis, maternal-zygotic mutant analysis, clonal analysis, rescue experiments) further demonstrates the rigor of the experiments performed here. While the scientific findings reported here may be primarily of interest to researchers who are interested in Talins and the coupling of integrin medicated cell adhesion to the actin cytoskeleton, I believe the approach that the authors are taking to study the mechanistic function of an important protein domain identified through biochemical studies will be more broadly applicable (e.g. functional studies of genetic variants identified in human diseases). I believe this is a strong manuscript that deserves to be published in PLoS Genetics without any major alterations. I would like to congratulate the authors for their achievement.

Major Issues: None

Minor Issues:

1) This paper will tremendously benefit of showing a schematic model diagram of how the authors think the two different actin binding domains in Talin works at the end of the paper (e.g. analogous to a graphical abstract). While the authors do a good job explaining how the genetic results should be interpreted in text, it would be great if they can provide a visual representation of a Talin (or Talins) working as a dimer and how the two domains are providing different functions, especially for researchers who are not in the Talin/Integrin/Actin field.

2) Line 93 and elsewhere: The authors sometimes italicize the word ‘Drosophila’ and sometimes they do not. They should be consistent (I personally recommend italicizing it everywhere).

3) Line 102: When the author say “flies have only one talin gene”, they should mention that this gene is called ‘rhea’ here. Otherwise, some readers may fail to make the connection between talin and rhea later in the paper.

4) Line 166 and line 169: URL for FlyCRISPR website does not have to be shown in the main text. It would be sufficient to refer to it in the Methods section, and you can just site the relevant manuscript here, which I believe is Ref 47.

5) Lines 227-227 and 232: ‘Retraction’, ‘Dorsal’ and ‘Closure’ do not have to be capitalized.

6) Line 260: ‘Cell’ doesn’t have to be capitalized here.

7) Line 459: ‘was’ should be ‘were’.

8) There may be several typos regarding amino acid numbers throughout the text.

Please check carefully when finalizing the work.

line 136, 144: 940(correct residue should be: 941), 941(942), 948(949)

line 140, 145, 159, 180: 2670(2450), 2671(2451), 2672(2452)

line 157, 176: 940D(941E), 941D(942E), 948D(949E)

Reviewer #2: This MS characterises the in vivo function of two actin binding sites in Talin, ABS2 and 3 by studying various phenotypes in crispr generated mutants in these domains. The findings suggest that these ABS have both unique and complementary roles in integrin mediated adhesion. They also provided support for talin acting as a dimer in Drosophila, supporting previous work in cells.

This is a clear, well presented, original work on trying to understand further the function of talin in integrin mediated adhesion, crucial for many morphogenetic processes. There phenotypic characterisation focuses on muscle-tendon attachments, but the MS also explore other events during embryogenesis or larvae behaviour.

I would like to raise a few points, and suggest a few experiments that may help the MS. One key aspect for the conclusions is that the binding capacity of the ABS is truly knocked out with the chosen mutations, especially for ABS3 since the weaker phenotypes of this mutant leads the arguments towards the unique versus collaborative work of the two ABS, so is the actin binding capacity of the ABS3 mutants truly gone in vivo? I think the authors should check this, maybe by ipps from embryos?

The authors say: ‘Using the approach detailed here we successfully generated multiple independent mutated lines of ABS2 and ABS3’…how many lines have been tested for the phenotypes described all long the MS? Is it just one line of each for all experiments?

For wings, DC and GBR, both ABS2 and 3 mutant phenotypes are similar to the null phenotype. For muscle attachment, then ABS2 is more important. This is again an important finding for the ‘unique versus collaborative’ hypothesis. But how are muscle attachments quantified? I couldn’t find it anywhere. It says muscles round up, but in some mutant embryos is difficult to see which one is attached or not (fig3c or d for example). How was the attachment of muscles phenotype quantified?

Part of the conclusion is that ABS3 is required ‘to maintain and strengthen’ integrins adhesion. How did the authors conclude this? Which experiments show this? I fail to conclude that. Yes, the phenotype is weaker than ABS2 and various proteins are clearly recruited in this mutant background, but that is not concluding that is required for maintaining or strengthening the complexes? Related to this, the rap1-binding mutant rheaK17E is a nice experiment and shows that ABS3 mutant proteins can recruit ABS2 mutant proteins when together in the cells. For strengthening this experiment, the authors should check rheaK17E levels of expression.

The authors say: ‘To test if these differences were due to differences in the expression of the ABS mutations, we carried out qRTPCR experiments which showed that both the ABS2 and ABS3 mutant…’ The authors should also check protein levels by western blot, as translation or stability of the protein could be different.

Why is the talin transgene not rescuing adult viability (fig2A)? This may be well known in the field, but it should be explained here too.

Can MHC levels also be quantified, as the authors did for F-actin? This measurement would contribute to understanding not just actin recruitment, but contractility capacity at the complex.

When studying the recruitment of complex components (fig6), how do the authors know where the MTJs are if they are not labelled with a 3rd component that is not affected in the mutant background? As in previous images, a marker such as integrins (can integrins be used?) should be used in combination with paxillin and pinch, so the MTJs can be identified and then the level of recruitment to those sites of paxillin and pinch can be measured.

Contrary to what the authors claim, I think vinculin has a substantial effect on viability, when looking at L1 survival rate (SR): compared wildtype larvae in fig2b (100% SR) with wildtype + vinculin in fig7B (68.2% SR), Similarly, larvae motility phenotype in ABS3/3 and ABS2/3 mutant is nicely rescued by vinculin, but I fail to see this for L1 SR, which is lower in fig7 than in fig2. This section needs to be more precise and accurate. This is important, as it may suggest that the role of vinculin-talin interaction may be different in different contexts.

In the vinculin-RFP experiment the flies have also endogenous vinculin, right? In this case, this experiment is looking at vinculin recruitment in conditions of vinculin overexpression. I think the authors need to do the same experiment without OE of vinculin, by antibody staining of endogenous one, or by an endogenous vinculin tagged.

Not an expert in the topic, but there are various relevant papers published in 2022 and 2023 that are not mentioned in this MS. (e.g., The C-terminal actin-binding domain of talin forms an asymmetric catch bond with F-actin, PNAS 2022. Owen et al.). The reference list needs updating, and more importantly, the findings need to be presented in the light of those papers, especially in the Discussion.

Minor point: the pinch antibody description is duplicated, but with different information, PINCH (rabbit, 1:1000; Mary Beckerle, University of Utah) or Mary Beckerle, Huntsman Cancer Institute, UT)

Reviewer #3: This manuscript by Camp and colleagues investigates the roles of the different actin binding sites in Talin for Talin function during Drosophila development. Talin has three putative actin binding sites (ABS) and can therefore link the actin cytoskeleton via the integrin tail, to which Talin binds, to the extracellular matrix. The authors tested the functional contribution of 2 of these actin binding sites, ABS2 and ABS3, by mutating several essential aa in each of them by Cas9 induced HDR. Functional tests identified an essential role for both of them, with ABS2 mutants dying as embryos or small L1 larvae, similar to Talin null alleles, and ABS3 mutants dying as L2 larvae, while trans-heterozygous combinations survive until pupal stage. Similarly, ABS2 mutants show stronger muscle detachment, germband retraction and dorsal closure phenotypes in the embryo than ABS3 mutants. Again, trans-heterozygous show only weak phenotypes.

Interestingly, ABS3 mutant Talin but not ABS2 mutant Talin is recruited to integrin at the muscle attachments, showing that the ABS2 site is essential for the initial recruitment of Talin to integrin. The same was found for actin enrichment at the MTJ.

Talin is proposed to act as a dimer since a long time. The authors substantiate this by combining the ABS2 mutant with a rescue construct that either has an ABS3 mutation or an ABS3 and a dimerization mutation. The latter does not rescue well suggestion that Talin dimer with ABS2/ABS3 mutations works better than single mutant proteins alone.

Overall, these data are interesting, they were nicely quantified, are of high quality and are well presented.

1. It would be interesting to better understand, why the ABS3 mutants die as L2 larvae, despite having only weak phenotypes in the embryo. The autho

---

## [Decision Letter · Decision Letter 1]

12 Mar 2024

Dear Dr Tanentzapf,

We are pleased to inform you that your manuscript entitled "The actin binding sites of talin have both distinct and complementary roles in cell-ECM adhesion" has been editorially accepted for publication in PLOS Genetics. Congratulations!

Yours sincerely,

Pablo Wappner

Academic Editor

PLOS Genetics

Gregory P. Copenhaver

Editor-in-Chief

PLOS Genetics

Comments from the reviewers (if applicable):

Reviewer's Responses to Questions

**Comments to the Authors:**

Reviewer #1: The authors sufficiently addressed my requests and recommendation. I believe the paper is suitable for publication in PLoS Genetics.

Reviewer #2: The authors have done an excellent work answering my raised questions.

Reviewer #3: The authors have clarified the few remaining concerns that I had both by new experiments or by better explanations in the text.

I now recommend acceptance of this interesting paper for publication in PLoS Genetics and want to congratulate the authors on their nice work.

**Have all data underlying the figures and results presented in the manuscript been provided?**

Reviewer #1: Yes

Reviewer #2: Yes

Reviewer #3: Yes

PLOS authors have the option to publish the peer review history of their article (what does this mean?). If published, this will include your full peer review and any attached files.

Reviewer #1: No

Reviewer #2: No

Reviewer #3: No

**Data Deposition**

http://datadryad.org/submit?journalID=pgenetics&manu=PGENETICS-D-23-01063R1

**Press Queries**

---

## [Editor Report · Acceptance letter]

17 Apr 2024

PGENETICS-D-23-01063R1 

The actin binding sites of talin have both distinct and complementary roles in cell-ECM adhesion 

Dear Dr Tanentzapf, 

We are pleased to inform you that your manuscript entitled "The actin binding sites of talin have both distinct and complementary roles in cell-ECM adhesion" has been formally accepted for publication in PLOS Genetics! Your manuscript is now with our production department and you will be notified of the publication date in due course.

With kind regards,

Zsuzsanna Gémesi

PLOS Genetics

On behalf of:
